# Global biogenic isoprene emissions 2013-2020 inferred from satellite isoprene observations

Hui Li[1,2#], Philippe Ciais[1], Pramod Kumar[1], Didier A. Hauglustaine[1], Frédéric Chevallier[1], Grégoire Broquet[1], Dylan B. Millet[3], Kelley C. Wells[3], Jinghui Lian[1,4], Bo Zheng[2,5]

[1]Laboratoire des Sciences du Climat et de l'Environnement, LSCE/IPSL, CEA-CNRS-UVSQ, Université Paris-Saclay, F-91191 Gif-sur-Yvette, France.

[2]Shenzhen Key Laboratory of Ecological Remediation and Carbon Sequestration, Institute of Environment and Ecology, Tsinghua Shenzhen International Graduate School, Tsinghua University, Shenzhen 518055, China.

[3]University of Minnesota, St. Paul, MN 55108, USA.

[4]Origins.earth, SUEZ Group, Immeuble Altiplano, 4 Place de la Pyramide, 92800 Puteaux, France

[5]State Environmental Protection Key Laboratory of Sources and Control of Air Pollution Complex, Beijing 100084, China.

*Correspondence to*: Hui Li (hui.li@lsce.ipsl.fr)

**Abstract.** Isoprene, the most emitted biogenic volatile organic compound, exerts a remarkable influence on atmospheric oxidation capacity, air quality, and climate. Most existing top-down atmospheric estimates of isoprene emissions rely on observational formaldehyde (HCHO) as an indirect proxy, even though HCHO is produced from multiple precursors. Recent advances in satellite retrievals of isoprene concentrations from the Cross-track Infrared Sounder (CrIS) enable a direct constraint on isoprene emission inversions. Yet global, multi-year isoprene-based atmospheric inversions are still lacking. Here, we present global, monthly biogenic isoprene emission maps spanning 2013–2020, derived from a mass-balance inversion framework that assimilates CrIS-retrieved isoprene columns into the LMDZ-INCA chemistry–transport model. The global biogenic isoprene emissions average is of $456 \pm 238$ TgC yr$^{-1}$ over 2013-2020, which is broadly consistent with existing inventories and HCHO-based inversion estimates. The LMDZ-INCA simulations using this estimate of the emissions exhibit improved spatial agreement and reduced biases relative to two independent satellite HCHO retrieval products and to ground-based optical measurements, confirming the robustness of this inversion framework. The seasonal cycle of emissions is dominated by the Northern Hemisphere, driven by the strong seasonality in temperature and vegetation biomes. Interannually, emissions vary by on average 14 TgC yr$^{-1}$ (1-sigma standard deviation). Two major emission peaks are found in 2015–2016 (456 TgC yr$^{-1}$) and 2019–2020 (478 TgC yr$^{-1}$), coinciding with El Niño and widespread extreme heat-wave events, underscoring the dominant influence of temperature anomalies that increase biogenic emissions. Regional analyses identify the Amazon as the largest contributor to the interannual variability, accounting for 22.3% of the global interannual variance in isoprene emissions. Temperature emerges as the primary driver of regional interannual emissions, with its influence modulated by leaf area index and radiation to varying degrees across regions. As one of the earliest attempts at a global, multi-year inversion based on isoprene

observations, this dataset provides input for air quality and climate-chemistry models. The isoprene emission
dataset is available at https://doi.org/10.5281/zenodo.16214776 (Hui et al., 2025).

## 1. Introduction

Isoprene (2-methyl-1,3-butadiene, $C_5H_8$), the most abundantly emitted biogenic volatile organic compound
(BVOC), accounts for 40%-60% of global BVOC emissions, with annual fluxes estimated between 350 and
600 TgC $yr^{-1}$, showing a considerable uncertainty (Sindelarova et al., 2022; Messina et al., 2016; Wang et al.,
2024a). Its emissions are primarily regulated by land cover type, leaf area, climate conditions (e.g.,
temperature, radiation), and atmospheric $CO_2$ concentration. Among these, land cover, global warming, and
rising $CO_2$ levels drive long-term emission trends, while extreme climate events govern short-term
fluctuations. Emission factors (EFs), defined as the rate of emissions per unit area under standardized light
and temperature conditions (Henrot et al., 2017), differ substantially among land cover types. Broadleaf trees
exhibit the highest EFs, followed by needleleaf trees, shrubs, grasses, and crops in decreasing order (Opacka
et al., 2021; Guenther et al., 2012). Recent studies further indicate that global warming can enhance isoprene
emissions from shrubs and sedges, highlighting their emerging role in biogenic fluxes (Wang et al., 2024d;
Wang et al., 2024b; Wang et al., 2024c). Of all climate variables, temperature is widely recognized as the
primary driver (Seco et al., 2022; Stavrakou et al., 2018), yet the variability of its influence across regions is
not well characterized. The role of $CO_2$ is nuanced: although $CO_2$ fertilization is estimated to have historically
enhanced isoprene emissions, future increases in $CO_2$ concentrations may suppress emissions through
physiological inhibition effects (Unger, 2013; Pacifico et al., 2012).
Once emitted, isoprene undergoes rapid atmospheric oxidation, primarily initiated by hydroxyl radicals (OH)
(e.g., ~1 h at [OH] = $5 \times 10^6$ molecules $cm^{-3}$ at T=298 K) and by ozone ($O_3$) (Bates and Jacob, 2019). Due to
its high reactivity, isoprene plays a pivotal role in tropospheric chemistry: it modulates the oxidative capacity
of the atmosphere, influences the atmospheric lifetime of greenhouse gases such as methane ($CH_4$) (Pound
et al., 2023; Zhao et al., 2025), and serves as a major precursor to secondary organic aerosols through
condensational growth and new particle formation, which exacerbate regional air pollution (Xu et al., 2021;
Curtius et al., 2024). Moreover, isoprene affects $O_3$ chemistry in a nonlinear manner—acting as a net source
under high-$NO_x$ conditions and a net sink in low-$NO_x$ regimes (Geddes et al., 2022). A similar $NO_x$
dependence is observed for formaldehyde (HCHO) yields from isoprene, where elevated $NO_x$ levels
accelerate production rates and increase the overall HCHO yield (Wolfe et al., 2016).
Accurately quantifying isoprene emissions is essential for improving air quality forecasts and climate-
chemistry model predictions. Two commonly adopted approaches are bottom-up models and top-down
atmospheric inversions. Among bottom-up models, the Model of Emissions of Gases and Aerosols from
Nature (MEGAN) is the most widely used. It parameterizes isoprene emissions as a function of climate
drivers such as light, temperature, and biological variables leaf area index (LAI) and phenology (Guenther et
al., 2012). Variability across inventories reflects both differences in parameterizing functional relationships

with climate drivers and, more importantly, inconsistencies in representing vegetation distributions, land-use changes, and EFs (Do et al., 2025; Messina et al., 2016). While improvements are ongoing, bottom-up estimates remain highly uncertain due to unclear EFs especially over tropical regions, structural limitations, and the complex physiological responses of plants to meteorological variability (Cao et al., 2021). Top-down inversion methods offer a complementary strategy by deriving emissions with atmospheric observations. Most existing inversions rely on satellite-retrieved HCHO, a major oxidation product of isoprene, and exploit the relationship between HCHO concentrations and isoprene fluxes (Millet et al., 2008; Barkley et al., 2013; Marais et al., 2012). However, HCHO-based inversions face inherent limitations, including the non-linear nature of isoprene–OH chemistry (Valin et al., 2016) which is also a challenge for isoprene-based inversions, uncertainties in $NO_x$-dependent HCHO yields, smearing effects causing spatial displacement between isoprene emissions and HCHO formation (Wolfe et al., 2016), and contributions from non-isoprene HCHO precursors such as $CH_4$ and other volatile organic compounds (Nussbaumer et al., 2021).

Direct atmospheric inversion assimilating isoprene concentrations provides a promising alternative to HCHO-based approaches, partly circumventing those limitations. Historically, this strategy was limited by the lack of atmospheric isoprene observations. Recent advances in infrared remote sensing now enable global retrievals of isoprene concentrations from satellites such as the Cross-track Infrared Sounder (CrIS) (Fu et al., 2019; Palmer et al., 2022; Wells et al., 2022), offering new opportunities for direct inversion. To date, however, isoprene-based inversions remain limited; to our knowledge, only a few studies have been conducted at the regional scale, focusing on areas such as the Amazon Basin, Asia, etc. (Sun et al., 2025; Wells et al., 2020; Choi et al., 2025). No global, multi-year continuous isoprene-based atmospheric inversion has been reported yet.

To fill this gap, we present a global, eight-year (2013–2020), monthly biogenic isoprene emission inversion, based on CrIS-retrieved isoprene concentrations derived through an artificial neural network (ANN) approach (Wells et al., 2020; Wells et al., 2022) and assimilated into the LMDZ-INCA 3D chemistry–transport model. This framework provides a direct top-down constraint on isoprene emissions, complementing traditional HCHO-based approaches and enabling the first global, multi-year assessment of isoprene fluxes. The inferred emissions capture key spatiotemporal patterns, including pronounced seasonal cycles dominated by the Northern Hemisphere and two major emission peaks in 2015–2016 and 2019–2020 linked to strong temperature anomalies. These advances highlight the sensitivity of biogenic emissions to temperature variability and demonstrate the potential of CrIS-based inversions to improve emission estimates. The resulting dataset provides a valuable resource for air quality forecasting and climate modeling, and offers valuable insights into biosphere–atmosphere interactions under changing environmental conditions.

## 2. Methods

### 2.1 Observations of isoprene and HCHO

This study employs three satellite datasets, CrIS isoprene, TROPOMI HCHO, and OMPS HCHO, along with
ground-based HCHO column observations from the Pandonia Global Network (PGN), to derive and evaluate
biogenic isoprene emissions. CrIS, a Fourier transform spectrometer aboard the Suomi National Polar-
orbiting Partnership (Suomi-NPP) launched on 28 October 2011, provides daily global observations around
13:30 local time (Han et al., 2013). We use global monthly-mean CrIS isoprene column concentrations from
January 2013 to December 2020 (resolution of 0.5° latitude × 0.625° longitude), retrieved using an ANN
approach that links spectral indices from CrIS radiances to isoprene columns based on a training dataset
constructed from an ensemble of randomized chemical transport model profiles (Wells et al., 2020; Wells et
al., 2022). As the ANN retrieval does not include scene-specific vertical sensitivity information, the CrIS-
retrieved isoprene columns are directly compared with model-simulated columns. It is noteworthy that CrIS
retrievals lack coverage in high-latitude regions north of 60°N (Fig. S1), where the inversion retains their
prior emission in this study.
Two independent satellite-based datasets of HCHO column concentrations, OMPS-NM and TROPOMI, are
used to indirectly evaluate the posterior-simulated HCHO columns. The instrument OMPS-NM, flown with
CrIS on Suomi-NPP, measures backscattered solar radiation in the 300–380 nm range at ~13:30 local time,
delivering near-global coverage with a spatial resolution of 50 km × 50 km (Abad, 2022; Nowlan et al., 2023).
We use its OMPS_NPP_NMHCHO_L2 retrieval dataset, applying standard quality filters:
main_data_quality_flag = 0, solar zenith angle (SZA) < 70°, and cloud fraction < 0.4. TROPOMI, a nadir-
viewing hyperspectral spectrometer aboard the European Sentinel-5 Precursor satellite launched in October
2017, provides global HCHO column densities at a similar overpass time (~13:30 local time), with finer
spatial resolution (7 km × 3.5 km prior to August 2019 and 5.5 km × 3.5 km thereafter). We use the
TROPOMI level 2 product (TROPOMI-RPRO-v2.4), filtered by qa_value ≥ 0.75 (ESA, 2020). To ensure
comparability with the satellite retrievals in evaluation, modeled HCHO concentrations from LMDZ-INCA
are first processed with the averaging kernels (AK) provided with the two satellite HCHO products to
generate respective model-equivalent columns, and then resampled to the satellite overpass times (~13:30
local time). All satellite datasets are regridded to a common spatial resolution of 1.27° latitude × 2.5°
longitude for consistency. The annual spatial distribution of the three satellite datasets over the globe is shown
in Fig. S1.
In addition to satellite data, we also incorporate ground-observed HCHO columns from the PGN network
(https://www.pandonia-global-network.org/) for independent evaluation of the posterior simulation of
HCHO concentrations. Considering data availability and consistency across all three HCHO datasets, we
select the year 2019 as a representative period for the posterior evaluation (Section 3.1).
**2.2 LMDZ-INCA global chemistry-transport model**
To establish the relationship between isoprene emissions and atmospheric concentrations, we use the LMDZ-
INCA global chemistry–aerosol transport model (Hauglustaine et al., 2004). The model is coupled with the
ORCHIDEE (Organizing Carbon and Hydrology in Dynamic EcosystEm) land surface model, which

dynamically simulates vegetation processes and provides prior estimates of biogenic isoprene emissions using the following formulation (Messina et al., 2016):

$$F = \text{LAI} \times \text{SLW} \times \text{EFs} \times \text{CTL} \times L \tag{1}$$

where LAI is the leaf area index, SLW is the specific leaf weight, EFs denotes the base emissions at the leaf level for a Plant Functional Type (PFT) at standard conditions of temperature (T=303.15 K) and photosynthetically active radiation (PAR=1000 µmol $m^{-2}$ $s^{-1}$), CTL is the emission activity factor representing environmental responses (e.g., to temperature and light), and $L$ accounts for leaf age-dependent modulation of emissions. A detailed description of the ORCHIDEE-based isoprene emission (global emissions: ~512 TgC $yr^{-1}$) scheme can be found in Messina et al. (2016). In addition to isoprene, ORCHIDEE also simulates emissions of other BVOC, including monoterpenes, methanol, acetone, sesquiterpenes, and others. A detailed comparison between ORCHIDEE- and MEGAN-simulated BVOC emissions is provided in Messina et al. (2016).

LMDZ-INCA contains a state-of-the-art $CH_4$–$NO_x$–CO–NMHC–$O_3$ tropospheric photochemistry scheme with a total of 174 tracers, including the chemical degradation scheme of 10 non-methane hydrocarbons (NMHCs): $C_2H_6$, $C_3H_8$, $C_2H_4$, $C_3H_6$, $C_2H_2$, a lumped C>4 alkane, a lumped C>4 alkene, a lumped aromatic, isoprene and α-pinene. The mechanism comprises 398 homogeneous, 84 photolytic, and 33 heterogeneous reactions, and is continuously updated to integrate newly identified chemical processes and reaction pathways, thereby improving the representation of atmospheric composition and oxidation capacity (Hauglustaine et al., 2004; Folberth et al., 2006; Pletzer et al., 2022; Sand et al., 2023; Terrenoire et al., 2022; Novelli et al., 2020; Wennberg et al., 2018). Reactions directly related to isoprene and HCHO are listed in Tables S1-S2. Global LMDz-INCA simulations are performed at a horizontal resolution of 1.27° latitude × 2.5° longitude, with 79 vertical hybrid sigma-pressure levels extending up to ~80 km, and are nudged to ERA5 wind fields. Turbulent mixing within the planetary boundary layer (PBL) is parameterized following Mellor and Yamada (1982) scheme while thermal convection is represented using the Tiedtke (1989) convection parameterization. The vertical profiles of LMDZ-INCA simulated isoprene and HCHO concentrations over Amazon region (Fig. S2) show a continuous decrease from the surface upward, consistent with previous studies (Fu et al., 2019; Hewson et al., 2015). Monthly global anthropogenic emissions of chemical species and gases are taken from the open-source Community Emissions Data System (CEDS) gridded inventories, wherein $NO_x$ emissions include eleven anthropogenic sectors and fertilizer-related soil sources, with global totals of around 113 Tg $yr^{-1}$ (Hoesly et al., 2018; Mcduffie et al., 2020). Fire emissions are taken from the Global Fire Emissions Database version 4 (GFED4) (Van Der Werf et al., 2017). For isoprene, monthly mean emissions from the input files are redistributed diurnally based on the local solar zenith angle to account for their strong photochemical dependence. Further details of the LMDZ-INCA configuration are provided by Kumar et al. (2025). A three-year spin-up simulation (2010–2012) is conducted to equilibrate the system, followed by a base simulation for 2013–2020. During the base simulation, isoprene and HCHO concentrations and isoprene emissions are sampled hourly. These hourly outputs are then used for model–observation comparisons and for performing the global inversion of isoprene emissions over the 2013–2020 period.

## 2.3 Inversion methodology

In order to assimilate CrIS isoprene retrievals into the LMDZ-INCA model, we apply the finite-difference mass balance (FDMB) inversion framework (Cooper et al., 2017). Given isoprene's short atmospheric lifetime, typically a few hours (~3 h at [OH] = $1 \times 10^6$ molecules cm$^{-3}$ at T=298 K) (Bates and Jacob, 2019; Fu et al., 2019), its horizontal transport is generally limited to a few tens of kilometers, supporting the assumption of a local relationship between emissions and column concentrations. Although this assumption may break down at high latitudes near the poles, its impact is negligible as isoprene emissions are largely confined to 60°S–60°N. In addition, in tropical regions with low NO$_x$, isoprene-driven OH suppression can prolong its lifetime and potentially violate the local linearity assumption (Wells et al., 2020). A detailed discussion of NO$_x$ effects is provided in Section 2.4. The final biogenic emissions for each model grid cell and month are calculated as follows:

$$E_{posterior,i,m} = E_{prior,i,m}(1 + \beta_{i,m} \frac{\Omega_{obs,i,m} - \Omega_{simu,i,m}}{\Omega_{simu,i,m}})$$

(2)

In Eq. (2), $i$ denotes the model grid cell in the 1.27° × 2.5° mesh, $m$ indicates the month, and $\Omega_{obs,i,m}$ and $\Omega_{simu,i,m}$ represent the observed and simulated monthly mean isoprene column concentrations (molecules cm$^{-2}$), respectively. To account for the strong diurnal variability of the isoprene column, $\Omega_{simu,i,m}$ only considers the CrIS overpass time (~13:30 local time) in its average, for consistency with $\Omega_{obs,i,m}$. $E_{posterior,i,m}$ and $E_{prior,i,m}$ refer to the posterior and prior isoprene emissions (kgC m$^{-2}$ s$^{-1}$), respectively. $\beta_{i,m}$ is a dimensionless factor representing the local relative response of modeled isoprene columns ($\Delta\Omega_{simu}/\Omega_{simu}$) to relative changes in prior emissions ($\Delta E_{prior}/E_{prior}$) as calculated below:

$$\beta_{i,m} = \frac{\Delta E_{prior,i,m} / E_{prior,i,m}}{\Delta\Omega_{simu,i,m} / \Omega_{simu,i,m}}$$

(3)

To derive $\beta_{i,m}$, we conduct two LMDZ-INCA simulations each year: one using the original ORCHIDEE-based prior isoprene emissions, and the other with those emissions uniformly reduced by 40% (based on the difference between simulated and observational isoprene columns). Sensitivity tests using alternative perturbations (+25%) confirm that $\beta_{i,m}$ is overall insensitive to the choice of perturbation magnitude, with global mean differences around -10% (average ($\beta_{+25\%}/\beta_{-40\%}$) ratio=0.9; Fig. S3). The robustness of $\beta$ is further discussed in Section 2.4. To avoid extreme changes, we keep $\beta_{i,m}$ within the range 0-10, and the inversion is performed only over land grid cells. An illustration of the spatial distribution of monthly mean $\beta$ values for 2019 is shown in Fig. S4, with a global annual mean of approximately 0.85. Lower $\beta$ values (around 0.6-0.7) are generally found over tropical hotspots such as the Amazon, while higher values (≥1) are found across much of the Northern Hemisphere, similar to previous studies (Wells et al., 2020). In this study, Posterior updates are only applied to grid cells with valid $\beta$ and CrIS observations, while emissions in the remaining grids are retained at their prior values. During 2013–2020, an average of 67.6% of land grid cells are updated per month, representing 99.0% of prior monthly emissions (Fig. S5), since missing data are concentrated in

high-latitude regions with low emissions. For a clearer regional analysis, we divide the globe into 15 regions,
as listed in Table 1 and shown in Fig. S6.
**Table 1**. Regional classification in this study, with classified map presented in Fig. S6.

| Abbreviations | Full names |
| --- | --- |
| AMZ | Amazon |
| RSAM | Rest of Southern America (other than Amazon) |
| EQAF | Equatorial Africa |
| NAF | Northern Africa |
| SEAS | Southeast Asia |
| CHN+KAJ | China+Korea+Japan |
| SAS | South Asia |
| SAF | Southern Africa |
| USA | The United States |
| MIDE | Mideast |
| OCE | Oceania |
| RUS+CAS | Russia+Central Asia |
| CAM | Central America |
| EU | Europe |
| CAN | Canada |

**2.4 The robustness of the linear relationship between isoprene concentrations and emissions**
A central assumption in our FDMB inversion framework is the linear response of isoprene concentrations to
changes in emissions within certain perturbations. To assess the robustness of this assumption, we identified
grids where the $\beta$ difference between the +25% and –40% perturbations is within ±20% (i.e., $\beta_{+25\%}/\beta_{-40\%}$ ratio
between 0.8 and 1.2 in Fig. S3). These grids account for 70.8% of global isoprene emissions, indicating that
the linearization approximately holds across most emissions in this study. The grid-scale statistics of $\beta_{+25\%}/\beta_{-40\%}$
shows that the average ratio falls within 0.86-0.90, and median value within 0.85-89 each month (Fig.
S7). The remaining deviations, primarily located in low-isoprene environments (Fig. S8), point to localized
nonlinear responses, yet the overall relationship between isoprene emissions and its concentrations can be
considered approximately linear at the grid scale within the range of perturbations and corrections of the
inversions. It is important to note, however, that the perturbation range (–40% to +25%) represents a
substantial 65% change in emissions, which may generate large deviations from linearity. In fact, emission
variations are typically moderate; in this study, more than 63% of the grid cells exhibit posterior–prior
differences within 65%, accounting for over 82% of the global total emissions on average, suggesting that $\beta$
is relatively insensitive to the magnitude of emission perturbations in most regions (Fig. S9).
To further asses the linearization, we take 2019 as an example year to apply the iterative finite difference
mass balance method following the approach of Cooper et al. (2017). After the initial inversion with a –40%
perturbation, subsequent iterations use a smaller –10% perturbation, as the first step already reduces the
model–observation bias substantially. The inversion is repeated using the updated emissions until
convergence, with the final solution obtained when the average model–observation differences across the 15
regions change by less than 5%. Convergence is achieved after four iterations. The comparison between the
single-step and four-iteration results shows that the global annual total emissions differ by about 5.3%, while
the largest regional difference occurs in Mideast (MIDE) at about -20% (Fig. S10). The iterative procedure
effectively reduces model–observation discrepancies, confirming the optimization capability of the inversion
system. However, given the relatively small difference from the single-step inversion and the high
computational cost, the single-step approach is considered sufficient for the long-term emission dynamics
analysis in this study.
Another sensitivity test excludes low-isoprene regions (two tests excluding grids with monthly mean columns
$<0.5 \times 10^{15}$ molec cm$^{-2}$ or $<1 \times 10^{15}$ molec cm$^{-2}$) from the inversion by keeping the prior unchanged, ensuring
that optimization occurs only where the linearization of the emission–concentration relationship is robust.
The resulting posterior shows minimal impact on global totals, with an annual difference of less than 9%
compared to the base inversion, and the largest regional deviation of about 40% occurring in Northern Africa
(NAF) and MIDE (Fig. S11). These results confirm that low-isoprene regions indeed contribute higher
uncertainties during optimization, consistent with the uncertainty assessment in Section 3.2. Nevertheless,
the interannual variability derived under this configuration remains consistent with that from the full
inversion, indicating that despite these uncertainties, the long-term emission dynamics identified in this study
are robust (Fig. S11).
**2.5 The impact of NO$_x$ concentration on inversion**
The linearity between isoprene concentrations and emission changes is strongly modulated by ambient NO$_x$
levels and by isoprene itself because both species directly influence the oxidative capacity of the atmosphere
and, consequently, the chemical lifetime of isoprene (Wennberg et al., 2018). Under high-NO$_x$ conditions,
isoprene oxidation proceeds efficiently due to rapid OH radical recycling, supporting a robust linear
relationship between concentrations and emissions.  In contrast, in low-NO$_x$ environments, the reduced
atmospheric oxidizing capacity prolongs the chemical lifetime of isoprene, leading to a superlinear response
where concentrations increase disproportionately with emissions (Fu et al., 2019; Wells et al., 2020). This
nonlinearity reduces the validity of the linear assumption in regions with low NO$_x$, necessitating a careful
evaluation of $\beta$ non-linearity and sensitivity to ambient NO$_x$ levels.
In the LMDZ-INCA simulations, NO$_x$ emissions are prescribed from the CEDS global inventories (Mcduffie
et al., 2020), which cover eleven anthropogenic sectors, including agriculture, energy production,
transportation (on-road and non-road), residential, commercial, and international shipping, as well as soil
NO$_x$ emissions from synthetic and manure fertilizers. Detailed configurations are provided in Kumar et al.
(2025). Compared to TROPOMI-retrieved NO$_2$ tropospheric columns from the TROPOMI-RPRO-v2.4
product, LMDZ-INCA simulates an overall negative bias, with NO$_2$ concentrations approximately 30% lower
than observed (Figs. S12–S13). This underestimation of NO$_2$ leads to an overestimation of isoprene lifetime
and, consequently, a systematic underestimation of $\beta$ in Eq. (3). The effect is particularly pronounced in
regions with high isoprene concentrations, consistent with the ~10% reduction of $\beta$ observed in the +25%
isoprene emission perturbation test (Fig. S2). To further assess the influence of $NO_x$ conditions on the
inversion, we perform a sensitivity test using +25% $NO_x$ emissions for 2019. The results show negligible
differences from the base inversion, with a global annual total deviation of less than 0.1% and the largest
regional difference of 0.9% over South Asia (SAS) (Fig. S14).
**2.6 The impact of prior choice on inferred isoprene emissions**
To evaluate the sensitivity of the inversion to the choice of prior emissions, two additional sensitivity
experiments are conducted using MEGAN-MACC (Sindelarova et al., 2014) and MEGAN-ERA5 (also
known as CAMS-GLOB-BIOv3.1) (Sindelarova, 2021; Sindelarova et al., 2022) isoprene inventories, both
of which are mechanistically distinct from the ORCHIDEE-based prior employed in the main analysis. The
inversions are performed for the year 2019 following the same setup and observational constraints. Results
show that the inferred global total isoprene emissions differ by less than 3.5% among the three prior
configurations: deviations between the MEGAN-MACC-based inversion (500 Tg yr$^{-1}$) and our posterior
global total (485 Tg yr$^{-1}$) are 3.1%, while those between the MEGAN-ERA5-based inversion (495 Tg yr$^{-1}$)
and our posterior are 2.1%, suggesting that the inversion framework remains robust to the choice of prior in
global annual totals (Fig. S15). From a regional perspective, the largest differences occur in Oceania, where
posterior emissions derived from MEGAN-MACC and MEGAN-ERA5 differ from our reference posterior
by 60.6% and 17.4%, respectively (Fig. S16). Although Oceania shows the largest posterior discrepancies
globally, these differences are substantially smaller than those in their priors (19 Tg yr$^{-1}$ in ORCHIDEE, 108
Tg yr$^{-1}$ in MEGAN-MACC, and 61 Tg yr$^{-1}$ in MEGAN-ERA5 in 2019), indicating that the inversion
effectively reconciles regional inconsistencies and converges toward observational constraints even where
prior emissions diverge markedly. Overall, these tests demonstrate that the optimized emissions are primarily
driven by observational constraints rather than by the characteristics of the prior inventory.
**3. Results**
**3.1 Evaluation of the posterior simulation of HCHO and isoprene**
As shown in Fig. 1, the posterior simulation improves over prior results, both in terms of spatial distribution
and correlation with observations. For HCHO, model grid-level comparison against TROPOMI retrievals
shows that the global Root Mean Squared Error (RMSE) decreases from $0.29 \times 10^{16}$ to $0.18 \times 10^{16}$ molecules
cm$^{-2}$, reflecting a substantial improvement in model–observation agreement relative to the prior simulation.
Similar improvements are seen when compared with OMPS HCHO retrievals (Fig. S17), indirectly
supporting the reliability of the posterior emissions. This enhancement is particularly pronounced over the
Amazon, where the RMSE decreases by $0.31 \times 10^{16}$ molecules cm$^{-2}$ (Fig. S18). For isoprene, the model–
observation agreement improves more substantially, validating the linearization of LMDZ-INCA based on a
perturbation and the assumed local relationship between emissions and column concentrations. The
regression slope between posterior simulations and CrIS observations decreases from 2.61 to 1.07, while

RMSE reduces from $5.69 \times 10^{15}$ to $1.22 \times 10^{15}$ molecules cm$^{-2}$. Biases in key tropical regions such as the Amazon are notably reduced, with regional RMSE of isoprene decreasing by $19.59 \times 10^{15}$ molecules cm$^{-2}$ (Fig. S18). In addition to satellite comparisons, posterior-simulated HCHO also shows a modest improvement in agreement with ground-based HCHO column concentrations from the PGN network, with the RMSE decreasing from $0.45 \times 10^{16}$ to $0.42 \times 10^{16}$ molecules cm$^{-2}$ (Fig. S19). In 2020, when more PGN sites became available (increasing from 15 in 2019 to 20), the posterior HCHO concentrations also better match the PGN observations, with the RMSE decreasing from $0.49 \times 10^{16}$ to $0.47 \times 10^{16}$ molecules cm$^{-2}$ (Fig. S19). These improvements relative to various HCHO observations consistently demonstrate the ability of the inversion framework to derive reliable estimates of the isoprene emissions and enhance model performance across diverse observational benchmarks.

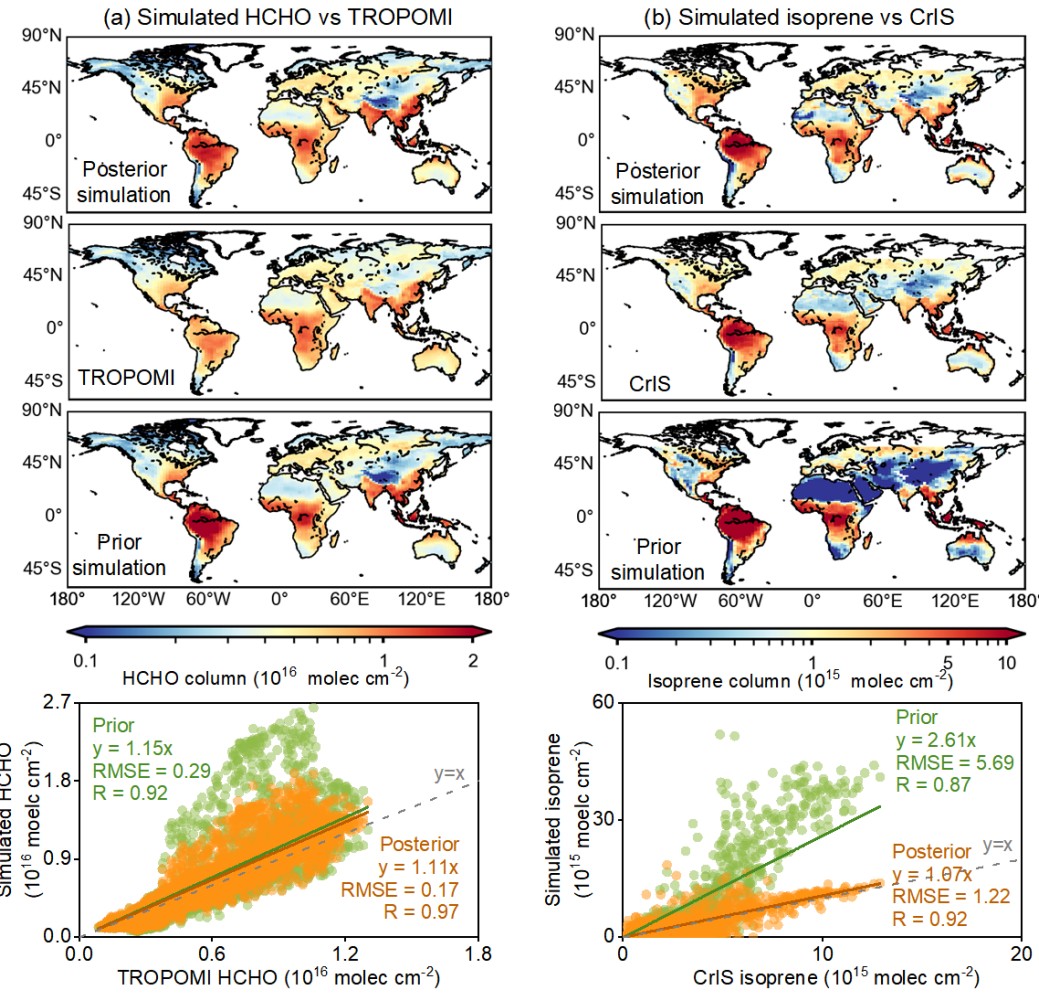

**Figure 1. Evaluation of the posterior LMDZ-INCA simulation using TROPOMI HCHO and CrIS isoprene observations in 2019.** (a) and (b) present the comparison of the simulated HCHO with TROPOMI observations, and of the simulated isoprene with CrIS observations, respectively. From top to bottom: the global distribution of model grid-scale annual mean of the posterior simulation, satellite observation (from TROPOMI in (a) column and from CrIS in (b) column), prior simulation of the column concentrations, and correlation between annual-mean simulation and observation across the model grid-cells covered by the observation.

## 3.2 Uncertainty estimation

In the FDMB inversion framework, posterior uncertainty ($\sigma_p$) is analytically estimated by minimizing the mass balance cost function, following the formulation of Cooper et al. (2017). It is important to note, however, that $\sigma_p$ does not account for potential structural errors in the LMDZ-INCA model, such as uncertainties in chemical mechanisms or meteorological fields. This limitation highlights the importance of independently evaluating the posterior estimates against external datasets to assess the robustness and reliability of the inferred emissions (seen in Section 3.1).

$$\frac{1}{\sigma_p} = \frac{1}{\sigma_a^2} + \frac{1}{\sigma_\varepsilon^2}$$

(4)

where $\sigma_a$ and $\sigma_\varepsilon$ represent the relative uncertainties in prior emissions and in the gridded monthly satellite observations, respectively. The prior emissions used in this study are derived from ORCHIDEE, a bottom-up, process-based model. Its uncertainties stem from factors including LAI, SLW, EFs, CTL, and $L$ (as shown in Eq. 1). PFT-dependent EFs vary substantially across different emission inventories, assigned a high uncertainty of 100% (Do et al., 2025; Weber et al., 2023). Among the remaining factors, LAI and the light-dependent fraction (LDF) that controls the CTL term are especially influential. According to Messina et al. (2016), the relative difference in LAI between the ORCHIDEE model and MODIS observations is approximately 50%. Therefore, we assign a 50% uncertainty to LAI, while a 20% uncertainty is applied to the remaining parameters. Applying standard error propagation for multiplicative variables yields a combined prior uncertainty ($\sigma_a$) of 117.0%, which represents a rough estimation of the overall uncertainty:

$$\sigma_a = \sqrt{\sigma_{LAI}^2 + \sigma_{SLW}^2 + \sigma_{EFs}^2 + \sigma_{LDF}^2 + \sigma_L^2}$$

(5)

The CrIS isoprene retrievals used in this study are based on an ANN retrieval approach. Retrieval uncertainties are spatially variable, depending on the column concentrations. According to Wells et al. (2022), retrieval uncertainties are generally <25% over high-concentration area ($\geq 10 \times 10^{15}$ molec cm$^{-2}$), and >50% in low-concentration area ($< 2 \times 10^{15}$ molec cm$^{-2}$). To account for this, we apply a piecewise uncertainty function for $\sigma_\varepsilon$ based on the observed isoprene column in each grid cell. An additional 20% uncertainty is applied to account for potential systematic effects, informed by the discrepancies observed in independent dataset comparisons (Wells et al., 2022). Here we assume these two uncertainty components (random retrieval error and systematic error) to be independent and additive in a simplified linear formulation, such that the final observational uncertainty is set at 45% for grid cells with $\Omega_{obs} \geq 10 \times 10^{15}$ molec cm$^{-2}$, varies linearly between 45% and 70% for $2 \times 10^{15}$ molec cm$^{-2} < \Omega_{obs} < 10 \times 10^{15}$ molec cm$^{-2}$, and the same linear relation is extrapolated to $\Omega_{obs} < 2 \times 10^{15}$ molec cm$^{-2}$ with an upper cap of 100%. Grid cells without valid observations remain at their prior values, and their posterior uncertainties are therefore set equal to the prior uncertainties. Prior and observational uncertainties are then combined using Eq. (4), and the resulting cell-level posterior relative uncertainties are aggregated to the global scale through area-weighted averaging. Taking 2020 as an example, the spatial distribution of cell-level posterior uncertainties is shown in Fig. 2, with the uncertainty for global annual isoprene emissions estimated at 51.6%.

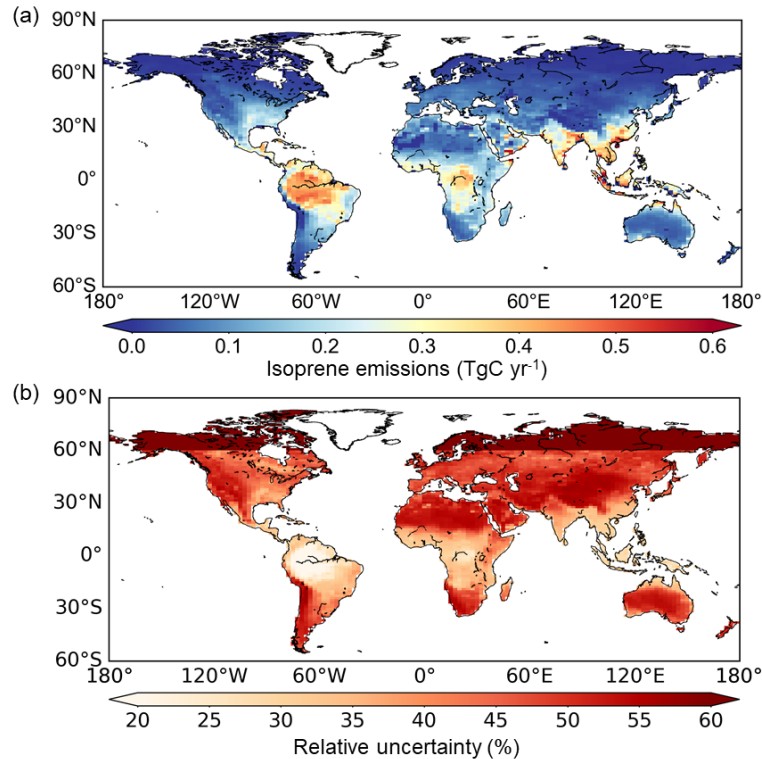

356

**Figure 2. (a) Global distribution of isoprene emissions (TgC per grid cell of 1.27° latitude × 2.5° longitude per year)**
**and (b) relative uncertainties (%) in 2020.** The uncertainties of global totals are area-weighted averages.

### 3.3 Seasonal pattern of isoprene emissions

Seasonally, the posterior emissions exhibit a pronounced peak during July–September (JAS), and a minimum
in December, January, and February (DJF) (Fig. 3). Over the study period (2013–2020), the global mean
monthly isoprene emission is approximately 38 TgC month$^{-1}$, rising by 42% to 54 TgC month$^{-1}$ during JAS
and declining sharply by 34% to 25 TgC month$^{-1}$ during DJF. This seasonal cycle agrees with recent HCHO-
based inversion results (Müller et al., 2024) but differs markedly from that in current bottom-up inventories:
MEGAN-MACC (Sindelarova et al., 2014) and MEGAN-ERA5 (also known as CAMS-GLOB-BIOv3.1)
(Sindelarova, 2021; Sindelarova et al., 2022) (Figs. 3 and S20). The discrepancy primarily stems from an
overestimation of isoprene emissions from Oceania (OCE) in current inventories. OCE contributes up to
92 TgC yr$^{-1}$ in MEGAN-MACC and 52 TgC yr$^{-1}$ in MEGAN-ERA5, exceeding half of the corresponding
emissions from the Amazon (AMZ, 103 and 94 TgC yr$^{-1}$, respectively), and exhibits substantial seasonal
variability (Fig. S21). Previous studies have attributed this likely overestimation of emissions and its
seasonality over OCE to the parameterization of temperature and radiation responses, along with the use of
high emission factors in bottom-up models (Emmerson et al., 2016; Emmerson et al., 2018). When OCE is
excluded, both MEGAN inventories show a JAS peak and DJF minimum, exhibiting a broadly similar
seasonal pattern to our posteriors (Fig. S22). Besides, sensitivity inversions using MEGAN-MACC and
MEGAN-ERA5 as priors also reproduce a JAS maximum and DJF minimum, reversing the original prior
seasonality. The posterior seasonality derived from all three priors aligns with that observed in CrIS isoprene

and OMPS HCHO concentrations (Fig. S20), indicating that the retrieved temporal variability reflects the
observed atmospheric signals and demonstrating the robustness of the inferred seasonal cycle.
The monthly variability in global isoprene emissions is largely driven by the Northern Hemisphere, mirroring
strong seasonal fluctuations in temperature (correlation coefficients, R=0.92) and vegetation activity (R with
LAI=0.89) (Figs. 3 and S23; Table S3). While these process relationships are inherently non-linear,
correlation analysis provides a useful first-order approximation of regional responses and sensitivities.
During JAS, Northern Hemisphere emissions peak at 41 TgC month$^{-1}$ and decline to 10 TgC month$^{-1}$ in DJF,
accounting for nearly ~100% of the global JAS–DJF peak-to-trough difference (~30 TgC). In contrast,
Southern Hemisphere emissions remain seasonally stable, averaging 14 TgC month$^{-1}$ during both JAS and
DJF with negligible difference. This strong hemispheric asymmetry underscores the dominant role of the
Northern Hemisphere in shaping the global seasonal cycle. Notably, the synchronicity between monthly
emissions and temperature is stronger in the Northern Hemisphere (R=0.96) than in the Southern Hemisphere
(R=0.54), reflecting the greater extent of mid-latitude land areas and sharper temperature seasonality in the
north (Figs. 3b-3c, and S24). Additionally, stronger LAI variations in the Northern Hemisphere further
reinforce this seasonal pattern (Figs. S25-S26) (Ren et al., 2024; Ma et al., 2023).

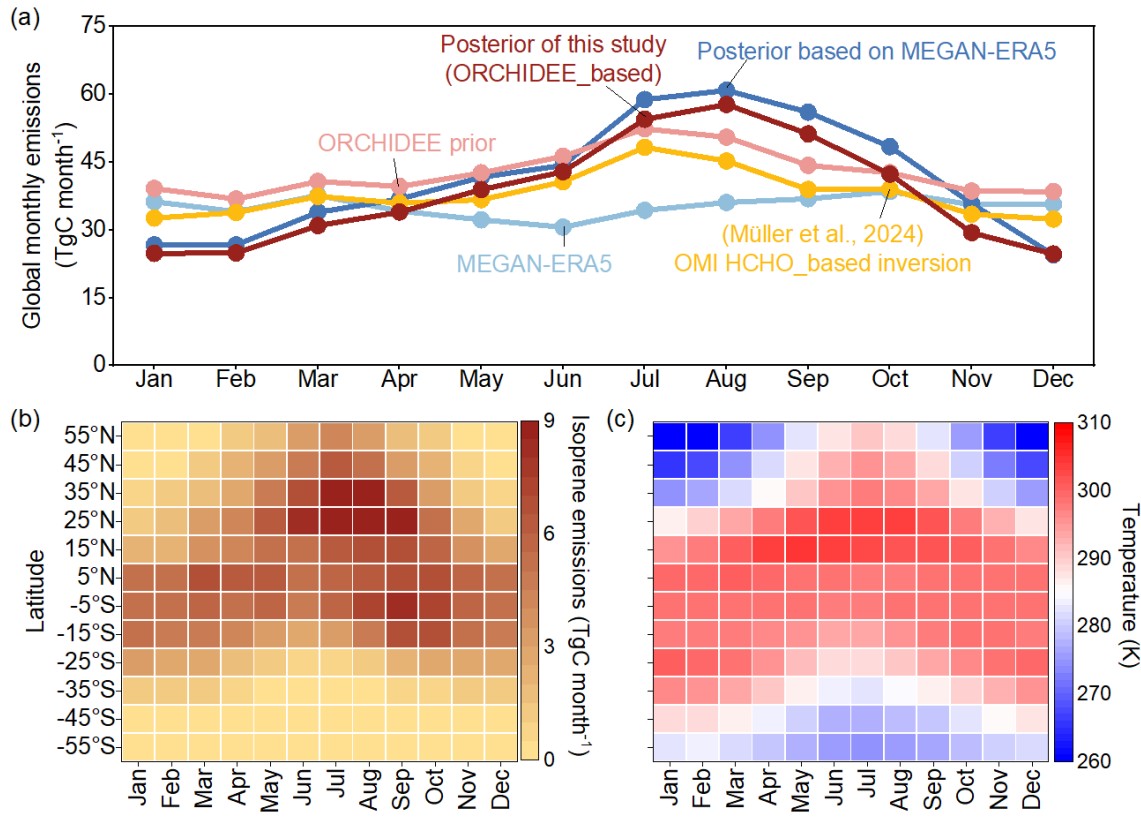


**Figure 3. Monthly mean isoprene emissions from 2013 to 2020.** (a) shows the global monthly pattern of ORCHIDEE
prior and our posterior in this study, MEGAN-ERA5 (also known as CAMS-GLOB-BIOv3.1) inventory (Sindelarova,
2021) and posterior based on MEGAN-ERA5, as well as OMI HCHO-based isoprene inversion result (Müller et al.,
2024). MEGAN-ERA5 is based on MEGAN v2.1, updated with ERA5 meteorology and CLM4 land cover (Sindelarova
et al., 2022). (b)-(c) display monthly distributions of our estimated isoprene emissions (TgC) and temperature (K) by
every 10° latitude band, respectively. We here only present the latitude range from 60°S to 60°N where emissions
dominate (~99%). Temperature is acquired from ERA5. The monthly distributions of two MEGAN inventories
(MEGAN-MACC and MEGAN-ERA5), precipitation from ERA5, and the Leaf area index (LAI) from Pu et al. (2024)
are presented in Fig. S25.

**3.4 Interannual variation of global isoprene emissions**

Over the study period (2013–2020), our global annual isoprene emissions average $456 \pm 238$ TgC yr$^{-1}$, falling
within the range of existing bottom-up inventories and satellite-based inversion estimates (Fig. 4; Tables S2–
S3). This value aligns closely with the MEGAN-ERA5 inventory (422 TgC yr$^{-1}$), whereas MEGAN-MACC
reports a notably higher estimate of 573 TgC yr$^{-1}$, reflecting a positive bias relative to both our results and
other datasets. Such overestimations in earlier MEGAN versions have been documented at global (Bauwens
et al., 2016) and regional scales (Kaiser et al., 2018; Gomes Alves et al., 2023).
In terms of interannual variability, global annual isoprene emissions exhibit a standard deviation (1$\sigma$) of
14 TgC yr$^{-1}$ over 2013–2020, corresponding to a coefficient of variation of 3.1%. Despite differences in
absolute magnitudes, the year-to-year variability simulated by both MEGAN inventories remains broadly
consistent with our inversion-based estimates (R=0.62–0.64 for annual emission rates). This temporal
coherence underscores the robustness of our posterior in capturing interannual variability. The spatial
distribution of interannual variability is highly uneven, with tropical regions such as the AMZ, Equatorial
Africa (EQAF), and SAS acting as the principal contributors. These regions show relatively large interannual
standard deviations (2–3 TgC yr$^{-1}$, coefficient of variation: 3.3%-7.6%), primarily due to their status as global
isoprene emission hotspots (Fig. 4b). On average, AMZ, EQAF, and SAS account for 15.5%, 11.5%, and
6.7% of global isoprene emissions, with corresponding emission intensities of 10, 6, and 6 gC m$^{-2}$ yr$^{-1}$,
respectively (Fig. 4c).
A positive and a negative anomaly are observed in the interannual variation of global isoprene emissions,
associated with the 2019–2020 extreme heat event and post-El Niño cooling in 2017, respectively,
highlighting temperature as the primary driver of year-to-year variability. During 2019–2020, annual
emissions averaged 478 TgC yr$^{-1}$, 1.5$\sigma$ above the 2013–2020 mean (456 TgC yr$^{-1}$), with 2019 alone reaching
485 TgC yr$^{-1}$ (2$\sigma$ above the mean) (Fig. S27). This peak coincides with widespread extreme heat (Robinson
et al., 2021), with elevated temperatures observed across most regions, except for certain arid and semi-arid
tropical zones such as NAF, SAS, and MIDE (Fig. S28). In contrast, emissions dipped to a minimum of
435 TgC yr$^{-1}$ in 2017 (1.5$\sigma$ below the mean), with a cooling following the extreme 2015–2016 El Niño event,
the most intense since 1950 (Hu and Fedorov, 2017). Although partially masked by the subsequent 2019–
2020 peak, the 2015–2016 El Niño also triggered an earlier emission enhancement, with global emissions
averaging 456 TgC yr$^{-1}$, exceeding the 2013–2018 baseline mean of 449 TgC yr$^{-1}$ (Fig. S27). During this
period, most regions except OCE experienced substantial warming, surpassed only by the more extreme heat
of 2019–2020 (Fig. S28). These two identified emission peaks in 2015–2016 and 2019–2020 are consistently
reflected in both bottom-up inventories, and satellite observations of HCHO and isoprene concentrations (Fig.
S29). Based on these dynamics, we classify the study period into four phases: Phase I: 2013–2014 (average:
447 TgC yr$^{-1}$); Phase II: 2015–2016 (456 TgC yr$^{-1}$); Phase III: 2017–2018 (445 TgC yr$^{-1}$); and Phase IV:
2019–2020 (478 TgC yr⁻¹), to enable clearer analyses and to isolate the distinct emission anomalies associated
with major climate events.

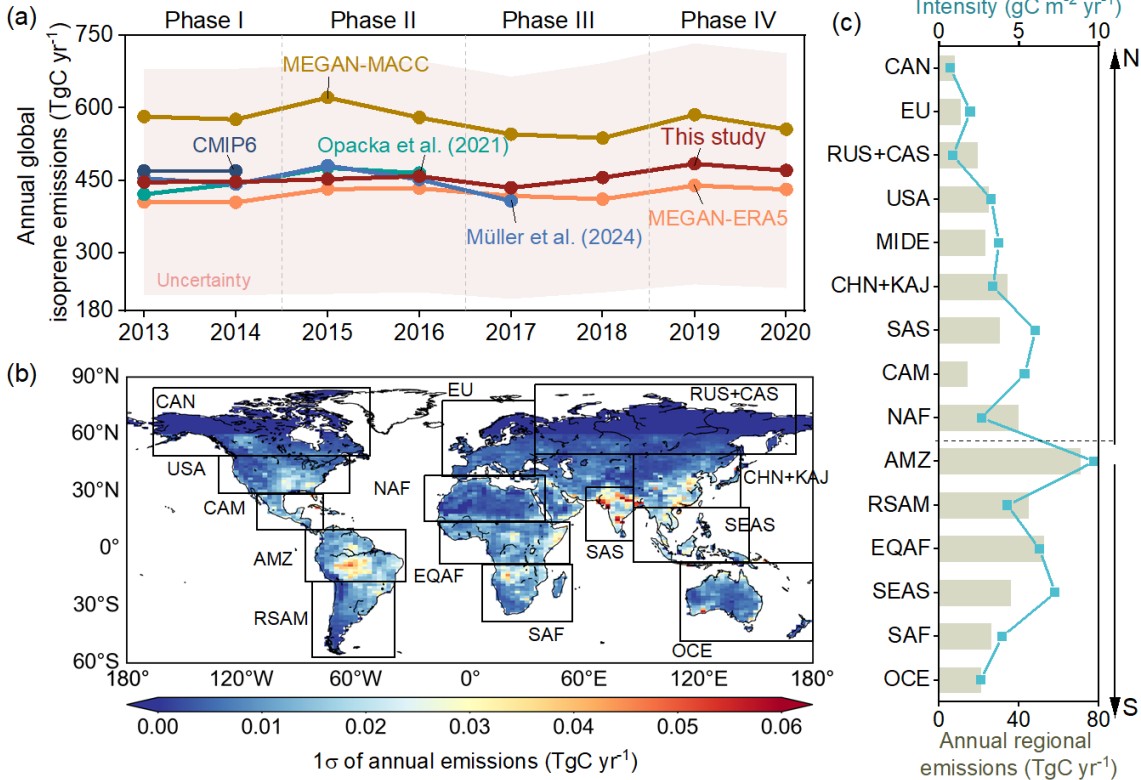

Canada (CAN) | Europe (EU) | Russia+Central Asia (RUS+CAS) | United States (USA) | Mideast (MIDE)
China+Korea+Japan (CHN+KAJ) | South Asia (SAS) | Central America (CAM) | Northern Africa (NAF) | Amazon (AMZ)
Rest of Southern America (RSAM) | Equatorial Africa (EQAF) | Southeast Asia (SEAS) | Southern Africa (SAF) | Oceania (OCE)

**Figure 4. Interannual isoprene emission variations from 2013 to 2020.** (a) compares the annual global isoprene emissions among the posterior (red shadow indicate the uncertainty), inventories including MEGAN-MACC, the MEGAN-ERA5 (also known as CAMS-GLOB-BIOv3.1) inventory, ensembles from Opacka et al. (2021), ensembles from CMIP6 (Do et al., 2025), and inversions based on corrected OMI HCHO observations (Müller et al., 2024). (b) plots the global spatial distribution of 1σ of annual isoprene emissions from 2013 to 2020, with frames corresponding to regions discussed in text. (c) depicts the regional annual emissions as well as the emission intensities (defined as the annual isoprene emissions per square meter per year). The regional classification is detailed in Fig. S6 of the SI and full names are listed below the figure.

## 3.5 Regional contribution to global interannual variations

Tropical regions emerge as the dominant drivers of interannual variability in global isoprene emissions, with the AMZ and RSAM identified as the largest contributors. From Phase I to IV, global emissions exhibit stepwise changes of +2.0%, –2.2%, and +7.2% relative to the preceding phase (Fig. 5a). Regional decomposition shows that the AMZ and RSAM together account for most of these changes: +7 TgC (80.9% of the global increase) during Phase I–II, –9 TgC (89.3% of the global decrease) during Phase II–III, and +9 TgC (27.7% of the global increase) during Phase III–IV. Their dominant influence reflects strong temperature sensitivity (9.0-25.5 TgC K⁻¹) (Figs. 5b-5d) and large interannual climate variability, particularly during the 2015–2016 El Niño, the following cooling, and the 2019–2020 heat events (Figs. S27–S28). The spatial patterns confirm this feature, with the largest emission fluctuations centered over the core Amazon

(Fig. 6a). Not all tropical regions exert such impacts on global interannual variations. EQAF and SEAS
display limited changes, contributing +1 TgC (7.5%) to the global increase during Phase I–II but offsetting
5.3% of the global decrease in Phase II–III with a net positive change of +1 TgC (Fig. 5a). This muted
response reflects regional heterogeneity in climate anomalies and ecosystem characteristics. EQAF,
dominated by grasslands (55.4%) and experiencing minor temperature anomalies during El Niño (Liu et al.,
2017), shows little emission change (Fig. 6b). In SEAS, widespread peatland fires in 2015 (Field et al., 2016),
likely triggered by extremely low precipitation (6.5 mm, 1.5σ below the mean; Fig. S32), may have
suppressed biogenic isoprene emissions in Phase II through vegetation loss and ecosystem disturbance
(Ciccioli et al., 2014). Both regions, however, exhibit emission increases in Phase IV, coinciding with
widespread warming (~1.0σ above their respective means) (Figs. 6b and S28).

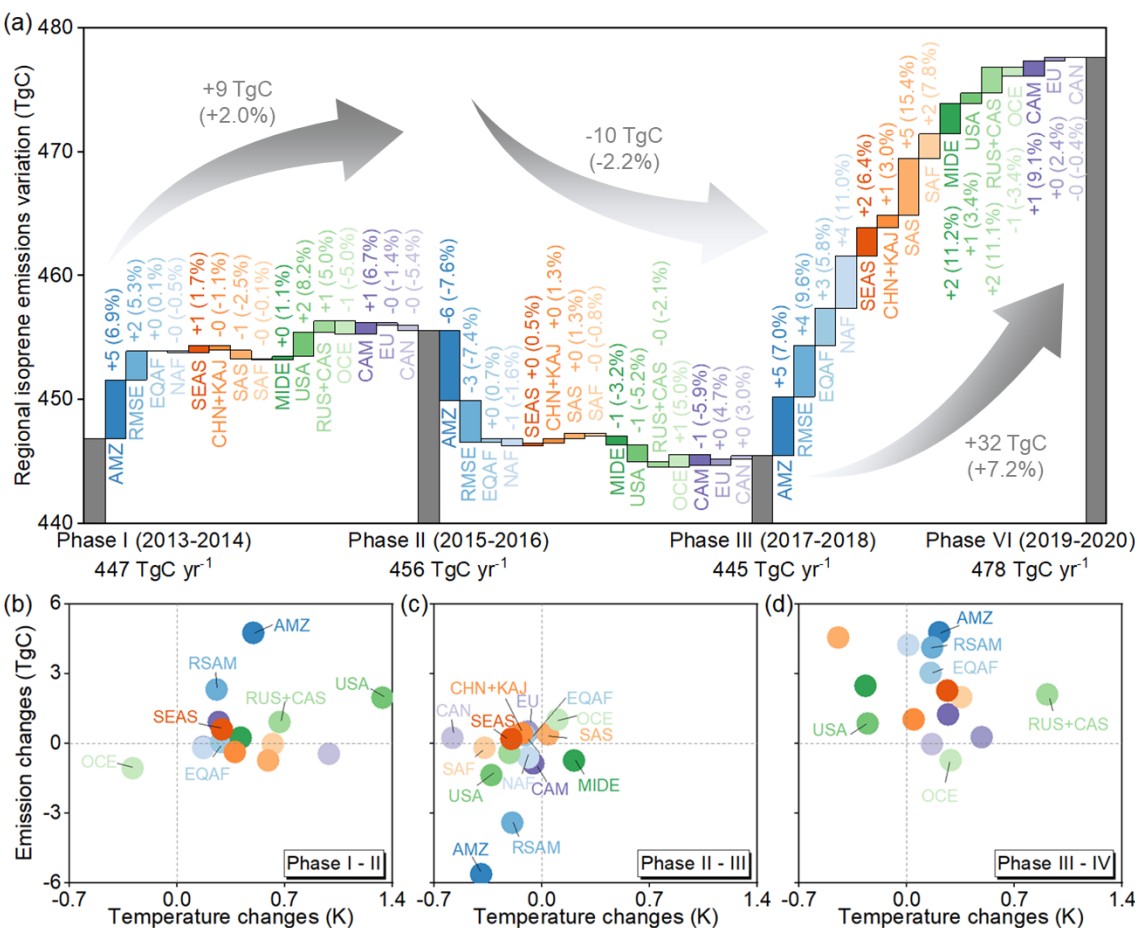


**Figure 5. Regional isoprene emission variations and meteorological changes over four phases.** (a) presents the
regional isoprene emission variation over four phases. (b)-(d) are the scatter plots between changes in regional isoprene
emissions and annual temperature from Phase I to II, II to III, and III to IV, respectively. (a)-(d) share the same legend,
with colors referring to different regions. Scatter plot of changes in regional isoprene emissions and Standardised
Precipitation-Evapotranspiration Index (SPEI), LAI, and radiation across phases are presented in Fig. S30.

Occasionally, non-tropical regions also contribute to the global interannual variability through extreme
anomalies. In the USA, emissions increased by 2 TgC (+8.2%) from Phase I to II, making it the third largest
contributor to the global increase during this period. In 2016, USA temperatures reached 285.8 K, 1.3σ above
its long-term mean (Fig. S28) and the highest warming observed among all regions during the 2015–2016 El
Niño. This temperature rise, coupled with enhanced LAI (+0.05) and stable hydrological conditions (Fig.
S30), favored increased photosynthetic activity and isoprene biosynthesis, elevating USA's contribution to
Phase II variability. The strong temperature sensitivity of USA isoprene emissions is consistent with previous
study (Abbot et al., 2003). Conversely, OCE stands out as an exception to the global trend with emission
changes of –1 TgC (–5.0%), +1 TgC (+5.0%), and –1 TgC (–3.4%). This pattern is linked to its temperature
changes (Figs. 5b-5d, S28, and S33), cooling during Phase II (–0.3 K and 1.1σ below its mean in 2016) and
subsequent temperature rebound (+0.1 K) in Phase III (Figs. 5c and 6c). The Phase IV decline is likely linked
to concurrent reductions in vegetation cover and intensified drought, particularly over northern Australia,
where LAI and SPEI decreased by around 0.1-0.2 and 0.5-1, respectively (Figs. S34–S35).

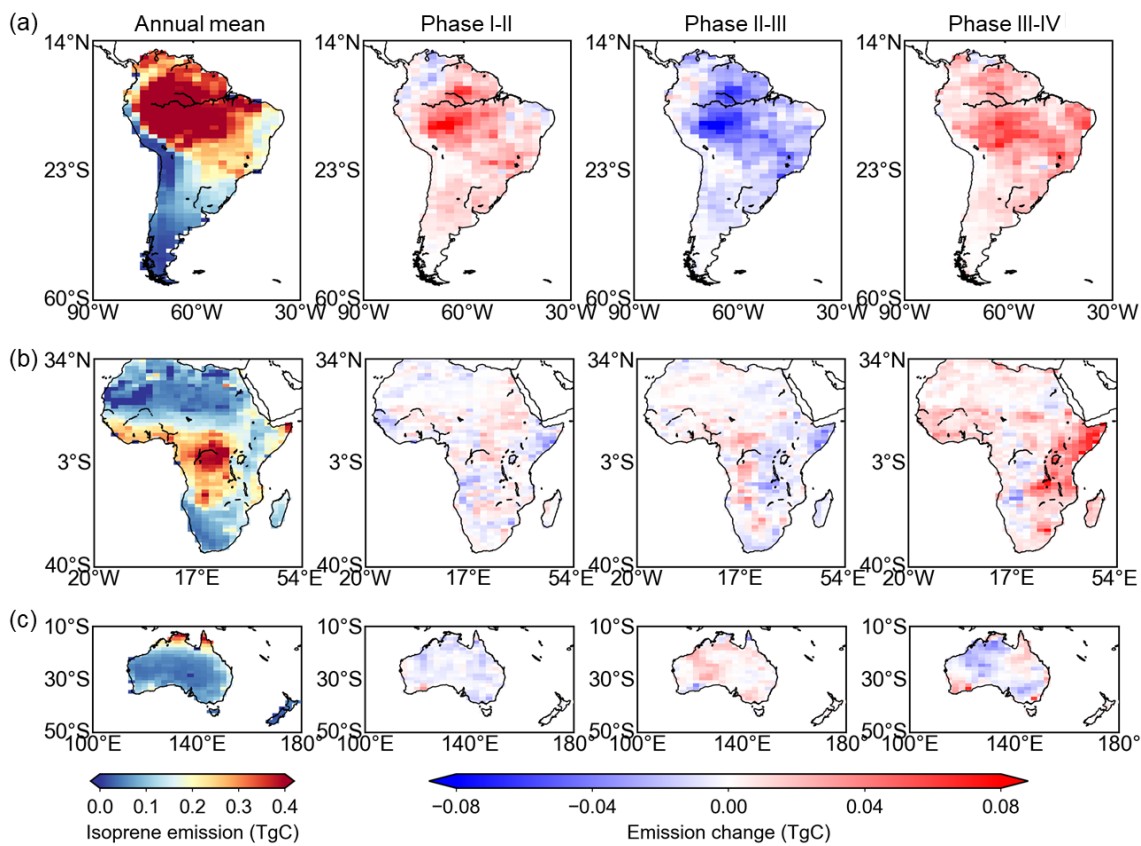


**Figure 6. Regional annual mean emissions and their changes across phases for (a) Southern America including AMZ and RSAM, (b) Africa including NAF, EQAF, and SAF, and (c) OCE.** The first column shows the annual mean isoprene emissions for each region, and the second to fourth columns correspond to the changes in regional isoprene emission across phases. Corresponding temperature, LAI, and SPEI distributions are shown in Figs. S33, S34, and S35.

**3.6 Drivers of regional isoprene emissions on a monthly scale**

To elucidate the underlying mechanisms and quantify regional sensitivities, we analyze R between monthly
isoprene emissions and key environmental variables, including temperature, solar radiation, LAI (Pu et al.,
2024), and drought index of Standardised Precipitation-Evapotranspiration Index (SPEI) (ECMWF,
2025), using both raw monthly values and monthly anomalies (calculated by removing the 2013-2020 mean
seasonal cycle for each month) (Figs. 7a–7b). To further assess whether temperature acts independently or
interacts with other factors, partial correlation analyses are performed (Figs. 7c–7d). Although biogenic
emission processes are inherently non-linear, these correlation analyses provide a useful first-order
approximation of regional sensitivities within the dynamic range observed in this study period.
Across most regions, isoprene emissions show strong positive correlations with temperature ($R > 0.5$, $p <$
$0.05$; Fig. 7a), suggesting temperature as the dominant first-order driver. Similar patterns are also observed
in the MEGAN-ERA5 inventory (Fig. S36). However, a notable difference appears in EQAF, where our
posterior results show no significant correlation with temperature, whereas MEGAN-ERA5 exhibits a strong
positive correlation. This finding is consistent with previous HCHO-based isoprene inversion studies, which
reported a reduced temperature dependence of isoprene emissions in the EQAF region (emission factor
decreased from 4.3 to 2.7 for evergreen broadleaf trees) (Marais et al., 2014). Partial correlation analysis (Fig.
7c) reveals that in many regions, including EU, MIDE, SAS, CAM, NAF, SEAS, and SAF, temperature
remains the primary independent driver of emissions (partial $R>0.5$, $p<0.05$). In contrast, in regions such as
CAN, USA, RUS+CAS, CHN+KAJ, AMZ, RSAM, and OCE, the temperature–isoprene relationships
weaken or become insignificant after controlling for other factors, suggesting that co-regulators such as
radiation and vegetation dynamics modulate this relationship. For example, in AMZ, the temperature–
isoprene correlation becomes insignificant when controlling for radiation (T|Rad, $p>0.05$), suggesting
radiation as a key co-regulator, consistent with the amplified temperature response observed in Phase IV.
EQAF presents a unique case: although no significant direct correlation with temperature is found, a positive
partial correlation emerges when controlling for LAI (T|LAI R=0.42), implying that vegetation dynamics
may obscure the underlying temperature sensitivity.
When monthly anomalies are used to isolate interannual variability (Figs. 7b and 7d), correlations between
temperature and isoprene emissions generally weaken, indicating that the strong monthly correlations largely
reflect seasonal co-variation. In most regions, temperature anomalies generally remain the dominant driver
($R>0$, $p<0.05$), albeit with weaker correlations than for the raw monthly values. Notably, AMZ (R=0.79) and
OCE (R=0.70) retain significant temperature–isoprene coupling, reflecting robust interannual temperature
sensitivity. In regions where temperature anomalies fail to explain interannual variability (e.g., SAS, CAN,
RUS+CAS), other drivers emerge. For instance, in SAS, LAI anomalies show the strongest association with
isoprene anomalies (R=0.65), underscoring the critical role of vegetation dynamics in controlling its
interannual emissions. Interestingly, in EQAF, where no significant correlation is found using raw monthly
data, temperature anomalies correlate significantly with isoprene anomalies, revealing an interannual
sensitivity previously masked by seasonal effects. Anomaly-based partial correlations further clarify the
independent role of temperature (Fig. 7d). Where direct correlations between temperature anomalies and
isoprene anomalies are significant, temperature generally remains an independent driver (partial $R>0$,
$p<0.05$), particularly in AMZ and OCE. In contrast, in regions such as CAN, RUS+CAS, and SAS, where
direct temperature–isoprene correlations are insignificant ($p>0.05$), interannual variability is dominated by
other factors. For example, in SAS, LAI anomalies exhibit the strongest (R=0.65 in Fig. 7b) and most

 independent association with isoprene anomalies, even after controlling for other variables (R=0.49-0.80),

underscoring the dominant role of vegetation dynamics in modulating interannual emissions in this region.

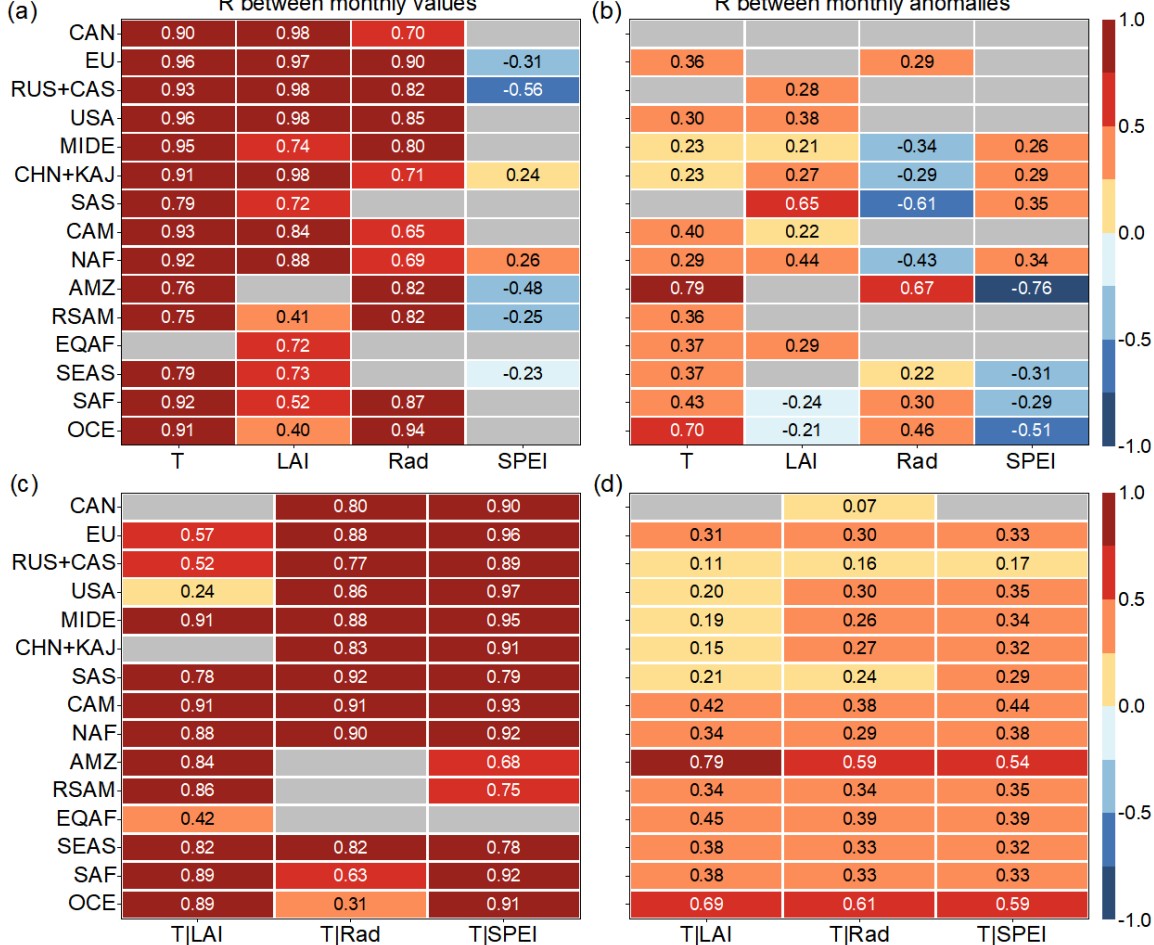


*T=Temperature; LAI=Leaf area index; Rad=Radiation; SPEI=Standardized Precipitation-Evapotranspiration Index.
(T|xx) indicates partial correlation between Temperature and Emissions after removing the influence of factor xx.

**Figure 7. Pearson correlation (R) matrix between regional isoprene emissions and environmental factors on a**
**monthly scale.** (a) shows the R matrix between monthly regional isoprene emissions and environmental factors. (c) plots
the partial correlation coefficient between temperature and isoprene emissions after removing certain factor's impact. (b)
and (d) are plotted for monthly anomalies obtained by removing the mean seasonal cycle as (a) and (c). In all panels, T
and Rad represent temperature and radiation, respectively. Regions are ordered from south to north (bottom to top). Gray
boxes indicate non-significant correlations (p>0.05).
Overall, temperature exerts the primary control on regional isoprene variability, but its apparent influence is
regionally modulated by co-varying environmental factors, particularly radiation and vegetation activity. The
anomaly-based correlations provide clearer evidence that while the apparent temperature dependence partly
reflects seasonal co-variation, interannual variability in tropical and subtropical regions is primarily governed
by coupled changes in temperature and ecosystem conditions.
**4. Limitations**

While our results demonstrate clear improvements over prior estimates in terms of both spatial distribution and correlation with observations (Figs. 1 and S17-S19), several limitations remain, highlighting areas for future refinement. A primary limitation arises from the incomplete spatial coverage of CrIS observations, particularly at high latitudes (north of 60°N; Fig. S1), where emissions in this study remain unchanged from prior. This omission has limited impact on global totals (~1.0% in prior), as boreal and tundra emissions are minor compared to tropical regions (Guenther et al., 2012). However, warming-driven increases in Arctic isoprene emissions (Seco et al., 2022; Wang et al., 2024d) suggest these regions may become more important in future global budgets and merit closer attention in upcoming inversions. Another limitation stems from comparing CrIS-retrieved isoprene columns with model outputs, as both are subject to uncertainties. The ANN-based retrieval lacks scene-specific vertical sensitivity information, introducing additional uncertainty in aligning the vertical profiles between observations and the model. Similarly, uncertainties in the LMDZ-INCA model's treatment of isoprene chemistry and transport may propagate into simulated columns. Moreover, relying on a single model framework introduces structural uncertainty that cannot be fully quantified here. Model-specific formulations of boundary layer mixing, photochemistry, and deposition can affect the simulated column–emission relationships. These issues could be mitigated through retrievals that include vertical sensitivity information, continued model development, and cross-model ensemble evaluations to better represent atmospheric isoprene processes.

Beyond satellite-related issues, several methodological constraints inherent to the inversion framework must be acknowledged. The FDMB approach assumes a localized linear relationship between surface emissions and atmospheric column concentrations, which simplifies the complex, non-linear chemistry of isoprene. This assumption is partly justified because CrIS observations are acquired near 13:30 local time, when OH concentrations peak and isoprene lifetimes are shortest (Hard et al., 1986; Karl et al., 2004). Moreover, this linearization is supported by sensitivity tests with varying perturbation magnitudes, increased $NO_x$ emission input, and improved posterior fits to CrIS observations. Nevertheless, the linearity between isoprene columns and emissions may break down across regions, especially in high-isoprene, low-$NO_x$ environment like the Amazon, where OH levels are limited (Zhao et al., 2025; Yoon, 2025). Future work could adopt joint $NO_x$–isoprene inversions or iterative schemes (Wells et al., 2020), to better capture the strong chemical coupling between $NO_x$, OH, and isoprene.

**5. Data and code availability**

All the data and model code are openly available. The isoprene emission data in this study are deposited in Zenodo (https://doi.org/10.5281/zenodo.16214776) (Hui et al., 2025). Other data include: the OMPS HCHO products are available in the NASA GES DISC for OMPS/Suomi-NPP (https://doi.org/10.5067/IIM1GHT07QA8); the TROPOMI HCHO products are available at https://sentiwiki.copernicus.eu/web/s5p-products; the 2013–2020 climatological means of the CrIS

isoprene columns are available at https://doi.org/10.13020/5n0j-wx73 (Wells et al., 2022). All the
meteorological factors (temperature, precipitation, and radiation) are acquired from ERA5 dataset at
https://cds.climate.copernicus.eu/datasets/reanalysis-era5-land-monthly-means?tab=overview. Land cover
data from 2013 to 2020 are ESA Land Cover Climate Change Initiative (Land_Cover_cci): Global Plant
Functional   Types   (PFT)   Dataset,   v2.0.8,   acquired   from
https://catalogue.ceda.ac.uk/uuid/26a0f46c95ee4c29b5c650b129aab788/. Pandonia Global Network (PGN)
surface observed HCHO area acquired from https://www.pandonia-global-network.org/. The drought indices,
i.e., the Standardised Precipitation-Evapotranspiration Index (SPEI), are obtained from ECMWF (https://xds-
preprod.ecmwf.int/datasets/derived-drought-historical-monthly?tab=overview). Leaf area index (LAI) data
are acquired from Pu et al. (2024). The codes and scripts developed for inversions, plotting, and other analysis
are accessible upon reasonable request from the corresponding author. The version of the LMDZ-INCA
model used in this study is available from: https://forge.ipsl.jussieu.fr/igcmg/svn/modipsl/trunk.
**6. Implication**
This study provides, to our knowledge, the first global, multi-year (2013–2020) estimates of isoprene
emissions derived directly from satellite-retrieved isoprene concentrations, offering valuable insights into the
temporal and spatial drivers of emission variability. Our analysis reveals the dominant influence of climate
anomalies in shaping both global and regional variability. On interannual timescales, two major emission
peaks in 2015–2016 and 2019–2020 coincide with El Niño and widespread extreme heat events, driven
primarily by temperature-induced enhancements in tropical regions, especially the Amazon. The elevated
biogenic isoprene emissions during the El Niño period are consistent with previous studies (Lathière et al.,
2006; Naik et al., 2004). Seasonally, global emissions peak during July–September (JAS) and reach a
minimum in December–February (DJF), reflecting the pronounced seasonality of temperature and vegetation
activity in the Northern Hemisphere. This seasonal pattern contrasts with the JAS minimum and DJF peak
simulated by the two MEGAN inventories. Sensitivity inversions using MEGAN-MACC and MEGAN-
ERA5 as priors yield consistent posterior seasonality, suggesting that bottom-up inventories likely
overestimate emissions in the Southern Hemisphere, especially over Oceania. Regarding temperature
sensitivity, MEGAN-based emissions generally display a more uniform response to temperature, whereas
our inversion indicates regionally differentiated sensitivities. For instance, in EQAF, temperature is not the
apparent dominant driver, implying that other factors, such as vegetation dynamics or solar radiation, exert a
stronger influence than represented in current models.
Given the sub-decadal scope of this study, the analysis has focused on short-term climate variability,
especially temperature, as the principal driver, while long-term influences such as land cover change and
rising atmospheric $CO_2$ concentrations are not explicitly addressed. Extending this framework to multi-
decadal periods will be essential to disentangle the interplay between short- and long-term drivers and to
assess their combined impacts on atmospheric chemistry and climate feedbacks. The occurrence of two major

climate anomalies, El Niño and widespread extreme heat events, supports the focus on extreme weather, which exerts disproportionate impacts on isoprene emissions. Looking ahead, however, the convergence of multiple environmental stressors, including global warming (Armstrong Mckay et al., 2022), deforestation in tropical regions (Leite-Filho et al., 2021), rising atmospheric $CO_2$ (with its dual fertilization and inhibition effects) (Cheng et al., 2022; Sahu et al., 2023), and the increasing frequency and intensity of climate extremes (wildfires, floods, and droughts) (Newman and Noy, 2023; Gebrechorkos et al., 2025; Zheng et al., 2023), raise critical questions about the long-term trajectory of global isoprene emissions. A key uncertainty is whether these interacting pressures will collectively amplify or suppress future emissions. Given isoprene's central role in regulating atmospheric oxidative capacity, such dynamics profoundly influence broader climate feedbacks. For instance, a sustained decline in isoprene emissions may elevate OH radical concentrations, thereby accelerating the atmospheric removal of $CH_4$ and other species (Zhao et al., 2025). However, the magnitude and direction of such feedbacks remain poorly constrained, highlighting the need for continued advancements in satellite observations and modeling tools to better characterize isoprene emissions and their interactions within the coupled biosphere–atmosphere system under future climate scenarios.

## Acknowledgements

This work was supported by the National Key R&D Program of China (Grant Nos. 2023YFC3709202), was granted access to the HPC resources of TGCC under the allocation A0170102201 made by GENCI, and was funded by ESA WORld EMission (WOREM) project (https://www.world-emission.com). We wish to thank J. Bruna (LSCE) and his team for computer support and the use of the OBELIX computing facility at LSCE, and thank Juliette Lathiere (LSCE) and her team for providing ORCHIDEE biogenic volatile organic compounds emissions. DBM and KCW acknowledge support from NASA GMAO (grant #80NSSC23K0520).

## Author contributions

HL designed this study, conducted the emission inversions, analyzed the data, and wrote the draft. PC, DH, BZ, and GB supervised the study, helped data analysis, reviewed and edited the paper. PK performed the LMDZ-INCA simulations, helped data analysis, and edited the paper. DB and KW offered the CrIS isoprene data, reviewed and edited the paper. FC and JL reviewed and edited the paper. All the co-authors contributed to the revision of this paper.

## Competing interests

At least one of the (co-)authors is a member of the editorial board of *Earth System Science Data*. The authors have no other competing interests to declare.

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
