# Peer review of "Global biogenic isoprene emissions 2013-2020 inferred from"

_Earth System Science Data, 2025_

## Author Comment (AC1)

This study established a mass-balance inversion framework that integrates CrIS-retrieved isoprene columns into the LMDZ-INCA chemistry-transport model to constrain global, monthly biogenic isoprene emission maps spanning 2013–2020. This represents a significant scientific contribution. Overall, I find the methodology generally sound, and particularly acknowledge the author' thorough justification of key assumptions and parameter selections. The manuscript is well-organized and clearly written. However, there are major concerns that must be addressed before I can recommend publication.

**Response:**

We sincerely thank the referee for their constructive and thoughtful comments on our manuscript. Below, we offer detailed responses addressing each point raised.

**Major Concern**

The validity of inversion critically depends on demonstrating whether posterior results converge when using different prior emissions inventories. The manuscript exclusively employs LMDZ-INCA land module simulations as the prior inventory. It is imperative to conduct additional inversions using an alternative emissions inventory—ideally one mechanistically distinct from LMDZ-INCA, with differing total magnitude, spatial distribution, and seasonal variation in isoprene emissions. This test would reveal whether the inferred total isoprene emissions align with those derived using LMDZ-INCA as prior, or quantify the extent to which discrepancies between the two prior inventories are reduced.

This validation using independent prior emissions is particularly crucial because the study finds that the posterior isoprene emission seasonal cycle using LMDZ-INCA as prior is entirely reversed compared to inventories such as MEGAN (Figure3). This is a new and important conclusion but has to be addressed carefully. What seasonal cycle does the prior LMDZ-INCA itself exhibit? Would using MEGAN as prior similarly yield a reversed seasonal pattern in the inversion? Without addressing these points, the scientific robustness of the posterior emissions—especially their seasonal cycle—remains limited. Given computational constraints, this sensitivity test could be restricted to a single representative year.

**Response:**

We have conducted sensitivity inversions using MEGAN-MACC and MEGAN-ERA5 as isoprene prior in 2019, respectively, which shows a minimal difference (<3.5%) in global annual posterior totals. Regionally, the largest difference occurs in Oceania, attributed to the substantial divergence in Oceania priors. For the seasonality, both MEGAN-MACC and MEGAN-ERA5 derived posteriors show a peak in JAS but reach minimum in DJF period, consistent with our findings. Besides, the satellite observed isoprene and HCHO column concentrations also exhibit a similar seasonal pattern as posteriors, which further demonstrates the reliability of isoprene posterior peak in JAS while minimum in DJF. Detailed discussion on sensitivity inversion on prior has been added as Section 2.6 The impact of prior choice on inferred isoprene emissions in the manuscript, and the aligned seasonality of the recent HCHO-based isoprene inversion, different prior tests, and satellite observations have been added in Lines 363-366, Lines 374-378, Figure 3, and S20.

**2.6 The impact of prior choice on inferred isoprene emissions**

To evaluate the sensitivity of the inversion to the choice of prior emissions, two additional sensitivity experiments are conducted using MEGAN-MACC (Sindelarova et al., 2014) and MEGAN-ERA5 (also known as CAMS-GLOB-BIOv3.1) (Sindelarova, 2021; Sindelarova et al., 2022) isoprene inventories, both of which are mechanistically distinct from the ORCHIDEE-based prior employed in the main analysis. The inversions are performed for the year 2019 following the same setup and observational constraints. Results show that the inferred global total isoprene

emissions differ by less than 3.5% among the three prior configurations: deviations between the MEGAN-MACC-based inversion (500 Tg yr-1) and our posterior global total (485 Tg yr-1) are 3.1%, while those between the MEGAN-ERA5-based inversion (495 Tg yr-1) and our posterior are 2.1%, suggesting that the inversion framework remains robust to the choice of prior in global annual totals (Fig. S15). From a regional perspective, the largest differences occur in Oceania, where posterior emissions derived from MEGAN-MACC and MEGAN-ERA5 differ from our reference posterior by 60.6% and 17.4%, respectively (Fig. S16). Although Oceania shows the largest posterior discrepancies globally, these differences are substantially smaller than those in their priors (19 Tg yr-1 in ORCHIDEE, 108 Tg yr-1 in MEGAN-MACC, and 61 Tg yr-1 in MEGAN-ERA5 in 2019), indicating that the inversion effectively reconciles regional inconsistencies and converges toward observational constraints even where prior emissions diverge markedly. Overall, these tests demonstrate that the optimized emissions are primarily driven by observational constraints rather than by the characteristics of the prior inventory.

**Lines 363-366:**

"This seasonal cycle agrees with recent HCHO-based inversion results (Müller et al., 2024) but differs markedly from that in current bottom-up inventories: MEGAN-MACC (Sindelarova et al., 2014) and MEGAN-ERA5 (also known as CAMS-GLOB-BIOv3.1) (Sindelarova, 2021; Sindelarova et al., 2022) (Figs. 3 and S20)."

**Lines 374-378:**

"Besides, sensitivity inversions using MEGAN-MACC and MEGAN-ERA5 as priors also reproduce a JAS maximum and DJF minimum, reversing the original prior seasonality. The posterior seasonality derived from all three priors aligns with that observed in CrIS isoprene and OMPS HCHO concentrations (Fig. S20), indicating that the retrieved temporal variability reflects the observed atmospheric signals and demonstrating the robustness of the inferred seasonal cycle."

**Figure 3**. Monthly mean isoprene emissions from 2013 to 2020. (a) shows the global monthly pattern of ORCHIDEE prior and our posterior in this study, MEGAN-ERA5 (also known as CAMS-GLOB-BIOv3.1) inventory (Sindelarova, 2021) and posterior based on MEGAN-ERA5, as well as OMI HCHO-based isoprene inversion result (Müller et al., 2024). MEGAN-ERA5 is based on MEGAN v2.1, updated with ERA5 meteorology and CLM4 land cover (Sindelarova et al., 2022). (b)-(c) display monthly distributions of our estimated isoprene emissions (TgC) and temperature (K) by every 10° latitude band, respectively. We here only present the latitude range from 60°S to 60°N where emissions dominate (~99%). Temperature is acquired from ERA5. The monthly distributions of two MEGAN inventories (MEGAN-MACC and MEGAN-ERA5), precipitation from ERA5, and the Leaf area index (LAI) from Pu et al. (2024) are presented in Fig. S25.

**Figure S20.** Monthly variation of isoprene emissions (ORCHIDEE prior and posterior, MEGAN-MACC prior and posterior, MEGAN-ERA5 and posterior) and CrIS observed isoprene column and OMPS HCHO column concentrations.

**Other Comments**

Sensitivity of perturbation magnitude (Lines 180–193):

The authors analyze the  $\beta+25\%/\beta-40\%$  value, with a global mean value of 0.9, to argue that the method is not sensitive to the perturbation magnitude and that the relationship between isoprene emissions and concentrations can be assumed linear. However, as seen in Figure S2,  $\beta+25\%/\beta-40\%$  ratios reveal substantial spatial heterogeneity. The manuscript should:

Present the grid-scale variance of  $\beta(+25\%)/\beta(-40\%)$  in the main text.

Further investigate how the nonlinearity reflected by  $\beta(+25\%)/\beta(-40\%)$  correlates with spatial patterns of OH, NO2 pollution, or other meteorological drivers.

**Response:**

We have added discussions on the grid-scale  $\beta_{+25\%}/\beta_{-40\%}$  ratio and its correlation with potential influential factors (isoprene, OH, NO2, and radiation) in Lines 219-224 in manuscript. Overall, the

statistics of grid-scale  $\beta_{+25\%}/\beta_{-40\%}$  shows that the average ratio falls within 0.86-0.90 across months, suggesting a localized linear-response exist in most grids. Grids with large-deviated  $\beta_{+25\%}/\beta_{-40\%}$  ratio occur in low-isoprene environments, where the linearity between isoprene emissions and concentrations weakens.

**Lines 219-224:**

"The grid-scale statistics of  $\beta_{+25\%}/\beta_{-40\%}$  shows that the average ratio falls within 0.86-0.90, and median value within 0.85-89 each month (Fig. S7). The remaining deviations, primarily located in low-isoprene environments (Fig. S8), point to localized nonlinear responses, yet the overall relationship between isoprene emissions and its concentrations can be considered approximately linear at the grid scale within the range of perturbations and corrections of the inversions."

**Figure S7**. Statistical summary of the grid-scale  $\beta$  ratio  $(\beta_{+25\%}/\beta_{-40\%})$  derived from the regression-based inversion in 2019. (a) shows the annual mean statistics, while (b)–(d) present the monthly distributions for Jan–Apr, May–Aug, and Sep–Dec, respectively. The tables inside (b)-(d) correspond to mean and median value of  $\beta_{+25\%}/\beta_{-40\%}$  for each month.

**Figure S8**. Correlation between annual mean grid-scale  $\beta$  ratio ( $\beta_{+25\%}/\beta_{-40\%}$ ) and (a) simulated isoprene column concentrations, (b) simulated OH column concentrations, (c) simulated NO2 column concentrations, and (d) radiation from ERA5.

**Model chemistry representation:**

The chemical mechanisms governing VOCs—especially isoprene oxidation pathways—vary significantly across models and directly influence simulated isoprene concentrations and their response to emissions. The model description section should explicitly detail the VOCs chemical mechanisms included in LMDZ-INCA.

The limitations section should explicitly address the potential impacts of using a single model framework on the results.

**Response:**

We have added a detailed description of the VOC chemical mechanisms, particularly those directly related to isoprene and HCHO in Lines 153-160 in the manuscript. In brief, LMDZ-INCA includes a comprehensive reactive VOC chemistry scheme with 14 isoprene reactions and 80 HCHO reactions, using up-to-date reaction rates.

In addition, we have included a discussion of the potential impacts of using a single model framework on the results in Lines 560–564 in the manuscript.

**Lines 153-160 in Section 2.2:**

"LMDZ-INCA contains a state-of-the-art CH4–NOx–CO–NMHC–O3 tropospheric photochemistry scheme with a total of 174 tracers, including the chemical degradation scheme of 10 non-methane hydrocarbons (NMHCs): C2H6, C3H8, C2H4, C3H6, C2H2, a lumped C>4 alkane, a lumped C>4 alkene, a lumped aromatic, isoprene and α-pinene. The mechanism comprises 398 homogeneous, 84 photolytic, and 33 heterogeneous reactions, and is continuously updated to integrate newly identified chemical processes and reaction pathways, thereby improving the representation of atmospheric composition and oxidation capacity (Hauglustaine et al., 2004; Folberth et al., 2006;

Pletzer et al., 2022; Sand et al., 2023; Terrenoire et al., 2022; Novelli et al., 2020; Wennberg et al., 2018). Reactions directly related to isoprene and HCHO are listed in Tables S1-S2."

**Lines 560-564 in Section 4:**

"Moreover, relying on a single model framework introduces structural uncertainty that cannot be fully quantified here. Model-specific formulations of boundary layer mixing, photochemistry, and deposition can affect the simulated column–emission relationships. These issues could be mitigated through retrievals that include vertical sensitivity information, continued model development, and cross-model ensemble evaluations to better represent atmospheric isoprene processes."

---

## Author Comment (AC2)

Review of "Global biogenic isoprene emissions 2013-2020 inferred from satellite isoprene observations" submitted by Hui Li et al.

This manuscript presents the first multiyear inversion of isoprene emissions based on spaceborne (CrIS) isoprene measurements and a global chemistry-transport model (LMDZ-INCA). The study is ambitious (probably too much) in that it aims to derive global gridded emissions at a high resolution (1.27 x 2.5 degrees) over an extended period (2013-2020); furthermore, it investigates in some detail the temporal variability of emissions and their correlation with meteorological and other variables. The high computational cost of emission inversion for a reactive species (known to strongly impact its own chemical lifetime through chemistry) is avoided through the use of a mass-balance approach, without iteration. The manuscript is generally well-written, and the topic is of great importance for the community. The CrIS dataset is a unique, and extremely valuable dataset for assessing the spatio-temporal variability of isoprene emissions. As expected, the results show that temperature is a major driving factor of isoprene temporal variability, while other factors (LAI, radiation, etc.) contribute as well.

Although to a large extent, the retrieved emissions are (at least qualitatively) validated by comparisons with formaldehyde datasets and by the analysis of temporal variability, I have several major reservations regarding the methodology used in this work (see below). In addition, some of the plots lack clarity (color bars, size) and important diagnostics are missing, which make it difficult for the reader to fully assess the method and the results. Finally, the results of this work should be better put in perspective with previous work, and the manuscript should cite the relevant literature when appropriate and better evaluate the results against previous work. My major comments are as follows.

**Response:**

We sincerely express our gratitude to the referee for constructive and insightful remarks regarding our manuscript. Below, we provide detailed responses addressing each point raised.

1) The assumption of linearity between emissions and column densities is not verified, despite the authors' claim that the issue is really minor. The slope of the relationship between emissions and columns (beta factor) is estimated from a reference run and a run using uniformly reduced emissions, by 40%. This is compared to an alternative estimation where the perturbed run uses increased emissions (+25%). The two estimations of beta differ by about 20% over much of the globe, in particular in July (Figure S2). Although not stated explicitly, the reason for adopting the case using decreased emissions (-40%) is motivated by the significant prior model overestimation against CrIS columns over rainforests, which account for a large fraction of the global emissions. Wherever the emission change deduced from CrIS is of the order of -40%, all is fine. But, Figure 1 shows many regions where emissions actually increase, and sometimes quite a lot. There, the optimized emissions are overestimated. Figure 1 displays many regions where the posterior model columns are significantly higher than CrIS, most notably Eastern and Central USA, southern China, the Middle East, and large parts of Canada, Europe and North Africa. Very probably, the emission enhancements are much larger than +25% at many of these places, and the linearity assumption breaks down. It would be easier to figure this out with a plot showing the ratio of posterior to prior model columns, not just annually but for different seasons, since the emission updates varies over time. It is impossible to tell from Figure 1 whether the -40% decrease is appropriate wherever CrIS suggests an emission decrease; at some locations, the decrease might be much larger than the -40% used in the beta estimation.

In their inversion of isoprene emissions based on CrIS and Geos-Chem, Wells et al. (2020) applied an iterative mass-balance approach, i.e. Equation 2 was applied iteratively "until convergence, with the final solution obtained when normalized model-measurement differences over isoprene hotspots change by <1%". The number of iterations needed for this criterion was not mentioned,

but it is safe to say that 3 is likely a minimum. I understand that iterative mass balance is more computationally demanding that the method used in this work, but I don't see the point of the high spatial resolution when the potential errors due to the method can be very large, and completely avoidable with the iterative approach. At the very least, iterative mass balance must be applied at least for one year, at least 3 or 4 iterations, so that the consequences might be assessed. I'm not even so much in favor of this option, because the issue might influence the interannual variability, at least over regions with low columns. For example, the interannual variability of retrieved emissions over India is higher than anywhere else (Fig. 4), probably due to a combination of large CrIS errors (due to low columns) and wrong emission optimization (due to non-linearity). To avoid such issues, areas with low columns could be simply left out of the optimization process.

**Response:**

We agree that the -40% perturbation used to derive the  $\beta$  does not apply uniformly across all regions. This perturbation magnitude was selected roughly based on the global annual mean difference between prior-simulated and CrIS-observed isoprene columns. To evaluate the validity of the linearity assumption and its potential influence on the inferred emissions and interannual variability, we performed additional sensitivity analyses: (1) an iterative finite-difference mass balance (IFDMB) inversion following Cooper et al. (2017) and Wells et al. (2020), using 2019 as a test year to assess the differences between single-step and iterative inversions; and (2) a low-isoprene exclusion test, in which grids with monthly mean isoprene columns below  $1 \times 10^{15}$  molec cm-2 or  $0.5 \times 10^{15}$  molec cm-2 were excluded from optimization to focus the inversion on regions with stronger linear relationships.

Results show that iterative inversions improve the fit between model and observations, confirming the optimization capability of the approach. Nevertheless, differences between the single-step and four-iteration inversions remain moderate, with the global annual total varying by about 5.3% and the largest regional deviation observed over the Mideast (approximately –20%). The low-isoprene exclusion test yields a global annual difference of less than 9%. These results indicate that, although localized non-linearities and uncertainties exist, particularly over low-signal regions, the inferred global and regional interannual dynamics remain robust. The detailed analyses and discussion of these tests have been added in Section 2.4 The robustness of the linear relationship between isoprene concentrations and emissions.

**Section 2.4 The robustness of the linear relationship between isoprene concentrations and emissions.**

A central assumption in our FDMB inversion framework is the linear response of isoprene concentrations to changes in emissions within certain perturbations. To assess the robustness of this assumption, we identified grids where the  $\beta$  difference between the +25% and -40% perturbations is within  $\pm 20\%$  (i.e.,  $\beta_{+25\%}/\beta_{-40\%}$  ratio between 0.8 and 1.2 in Fig. S3). These grids account for 70.8% of global isoprene emissions, indicating that the linearization approximately holds across most emissions in this study. The grid-scale statistics of  $\beta_{+25\%}/\beta_{-40\%}$  shows that the average ratio falls within 0.86-0.90, and median value within 0.85-89 each month (Fig. S7). The remaining deviations, primarily located in low-isoprene environments (Fig. S8), point to localized nonlinear responses, yet the overall relationship between isoprene emissions and its concentrations can be considered approximately linear at the grid scale within the range of perturbations and corrections of the inversions. It is important to note, however, that the perturbation range (-40% to +25%) represents a substantial 65% change in emissions, which may generate large deviations from linearity. In fact, emission variations are typically moderate; in this study, more than 63% of the grid cells exhibit posterior-prior differences within 65%, accounting for over 82% of the global total emissions on average, suggesting that  $\beta$  is relatively insensitive to the magnitude of emission perturbations in most regions (Fig. S9).

To further asses the linearization, we take 2019 as an example year to apply the iterative finite difference mass balance method following the approach of Cooper et al. (2017). After the initial inversion with a –40% perturbation, subsequent iterations use a smaller –10% perturbation, as the first step already reduces the model—observation bias substantially. The inversion is repeated using the updated emissions until convergence, with the final solution obtained when the average model—observation differences across the 15 regions change by less than 5%. Convergence is achieved after four iterations. The comparison between the single-step and four-iteration results shows that the global annual total emissions differ by about 5.3%, while the largest regional difference occurs in Mideast (MIDE) at about -20% (Fig. S10). The iterative procedure effectively reduces model—observation discrepancies, confirming the optimization capability of the inversion system. However, given the relatively small difference from the single-step inversion and the high computational cost, the single-step approach is considered sufficient for the long-term emission dynamics analysis in this study.

Another sensitivity test excludes low-isoprene regions (two tests excluding grids with monthly mean columns  $<0.5 \times 10^{15}$  molec cm-2 or  $<1 \times 10^{15}$  molec cm-2) from the inversion by keeping the prior unchanged, ensuring that optimization occurs only where the linearization of the emission—concentration relationship is robust. The resulting posterior shows minimal impact on global totals, with an annual difference of less than 9% compared to the base inversion, and the largest regional deviation of about 40% occurring in Northern Africa (NAF) and MIDE (Fig. S11). These results confirm that low-isoprene regions indeed contribute higher uncertainties during optimization, consistent with the uncertainty assessment in Section 3.2. Nevertheless, the interannual variability derived under this configuration remains consistent with that from the full inversion, indicating that despite these uncertainties, the long-term emission dynamics identified in this study are robust (Fig. S11).

**Figure S10.** Comparison between base inversion and sensitivity inversion with four-time iterations. (a) presents the global distribution of monthly difference in isoprene posteriors, and (b) compares the regional annual isoprene posteriors.

**Figure S11.** Comparison between base posteriors and sensitivity tests excluding grids of low isoprene columns, less than  $1x10^{15}$  molec cm-2 and  $0.5x10^{15}$  molec cm-2, respectively. (a) shows the global annual posteriors, and (b) present the one standard deviation (1 $\sigma$ ) of regional annual emissions from three posteriors from 2013 to 2020.

2) The LMDZ-INCA is not appropriately described. As far as I can tell from previous papers, the isoprene degradation mechanism was described by Folberth et al. (2006) and was based on earlier work (1999). Obviously, it does not incorporate the numerous mechanistic updates (e.g. OH recycling) prompted by laboratory and theoretical studies since then, of special importance at low-NOx (see e.g. Wennberg et al. 2018; Novelli et al. 2020; etc.). The consequences for the prediction of OH levels and HCHO formation are difficult to tell, but could be very large. This should be investigated, e.g. using a box model, to assess the performance of the LMDZ-INCA mechanism, in comparison with more recent ones. In absence of recycling mechanisms, the OH levels might be too low in the model at low NOx, leading to substantial overestimation of isoprene columns.

**Response:**

The chemical mechanism in LMDZ-INCA has been continuously expanded and updated over the past two decades and now includes OH recycling processes. The oxidation processes of isoprene by OH, NO3, and O3 have been added to LMDZ-INCA. A detailed description of the updated VOC chemistry, particularly for isoprene and HCHO (involving 14 and 80 reactions, respectively), has been added in Lines 153–160 in the manuscript, with reaction listings provided in Tables S1–S2.

**Lines 153-160 in manuscript:**

"LMDZ-INCA contains a state-of-the-art CH4–NOx–CO–NMHC–O3 tropospheric photochemistry scheme with a total of 174 tracers, including the chemical degradation scheme of 10 non-methane hydrocarbons (NMHCs): C2H6, C3H8, C2H4, C3H6, C2H2, a lumped C>4 alkane, a lumped C>4 alkene, a lumped aromatic, isoprene and α-pinene. The mechanism comprises 398 homogeneous,

84 photolytic, and 33 heterogeneous reactions, and is continuously updated to integrate newly identified chemical processes and reaction pathways, thereby improving the representation of atmospheric composition and oxidation capacity (Hauglustaine et al., 2004; Folberth et al., 2006; Pletzer et al., 2022; Sand et al., 2023; Terrenoire et al., 2022; Novelli et al., 2020; Wennberg et al., 2018). Reactions directly related to isoprene and HCHO are listed in Tables S1-S2."

3) A description of NOx and reactive VOC emissions should be provided, given the importance of NOx for OH levels (and hence isoprene) and VOCs for HCHO. Figure S6 suggests an underestimation of NO2 modelled tropospheric columns in comparison to TROPOMI, at least over tropical regions. However, the relative underestimation over key regions (e.g. Amazonia) is impossible to tell. This is essential to figure out, given the role of NOx for isoprene emission inversions (Wells et al. 2020). Note that Figure S6 shows features (red areas over Patagonia and parts of Australia) that are almost impossible to understand, and make me wonder whether the tropospheric column is correctly calculated.

Another model aspect requiring more information is PBL mixing. How does the model perform for the vertical profile of reactive species such as isoprene or similar compounds? This is relevant to model comparison with CrIS, because of the vertical dependence of the sensitivity of the instrument (Wells et al. 2020).

**Response:**

The NOx emissions are from CEDS inventory, and all the reactive VOC emissions are from ORCHIDEE (Organizing Carbon and Hydrology in Dynamic EcosystEm) land surface model. The BVOC emissions produced from ORCHIDEE model include many other species, like monoterpenes, methanol, acetone, sesquiterpenes, etc... We have added some description of BVOC and NOx emissions in Lines 149-152 (BVOC) and Lines 167-170 (NOx) in the manuscript.

For the original abnormal phenomenon in Fig. S12 (original Fig. S6), we made a visualization mistake: the latitude array was inverted. The data file is ordered from –90 to 90, but it was plotted from 90 to –90, which produced the artifact. Thank you for pointing this out, and we have corrected this figure.

Compared to TROPOMI observed  $NO_2$  (TROPOMI-RPRO-v2.4), LMDZ-INCA exhibit an overall underestimation (~30% lower), which could be attributed to lower  $NO_x$  emission input or the chemistry uncertainty in the model. To assess the  $NO_2$  concentration impact on the inferred isoprene emissions, we have conducted a sensitivity test by increasing  $NO_x$  emission input by 25% in 2019, which show a closer  $NO_2$  between LMDZ-INCA simulation and TROPOMI observation (Fig. S13). In this inversion, the global annual total exhibit less than 1% difference from base inversion, with largest regional deviance in South Asia (SAS). We have added the result of this sensitivity inversion in Lines 270-273.

Turbulent mixing within the planetary boundary layer (PBL) is parameterized following Mellor and Yamada (1982) scheme while thermal convection is represented using the Tiedtke (1989) convection parameterization. We have added this information in Lines 163-164.

We have plotted the vertical profile of simulated isoprene and HCHO over Amazon region as Fig. S2, which exhibit a continuous decrease from the surface upward, consistent with previous studies. We have added this description in Lines 165-167.

**Lines 149-152:**

"In addition to isoprene, ORCHIDEE also simulates emissions of other BVOC, including monoterpenes, methanol, acetone, sesquiterpenes, and others. A detailed comparison between ORCHIDEE- and MEGAN-simulated BVOC emissions is provided in Messina et al. (2016)."

**Lines 167-170:**

"Monthly global anthropogenic emissions of chemical species and gases are taken from the open-source Community Emissions Data System (CEDS) gridded inventories, wherein NOx emissions include eleven anthropogenic sectors and fertilizer-related soil sources, with global totals of around 113 Tg yr-1 (Hoesly et al., 2018; Mcduffie et al., 2020)."

**Lines 270-273:**

"To further assess the influence of  $NO_x$  conditions on the inversion, we perform a sensitivity test using +25%  $NO_x$  emissions for 2019. The results show negligible differences from the base inversion, with a global annual total deviation of less than 0.1% and the largest regional difference of 0.9% over South Asia (SAS) (Fig. S14)."

**Lines 163-164:**

"Turbulent mixing within the planetary boundary layer (PBL) is parameterized following Mellor and Yamada (1982) scheme while thermal convection is represented using the Tiedtke (1989) convection parameterization."

**Lines 165-167:**

"The vertical profiles of LMDZ-INCA simulated isoprene and HCHO concentrations over Amazon region (Fig. S2) show a continuous decrease from the surface upward, consistent with previous studies (Fu et al., 2019; Hewson et al., 2015)."

4) The analysis of results is long and often repetitive, and it does not cite properly the literature. Many findings are presented as new, while they were perfectly well known from past studies. Those studies should be cited and feed the discussion. Examples: the role of meteorological variables, especially temperature, is incorporated in emission models such as MEGAN, and has been verified using satellite measurements, see e.g. the Geos-Chem studies (e.g. Abbot et al. 2003) and Stavrakou papers (e.g. Stavrakou et al. 2018). The impact of El Nino on emissions was shown e.g. by Naik et al. (2004), Lathiere et al. (2006) and others.

The inversion results should be better evaluated against relevant literature. The ORCHIDEE emissions, being used as prior inventory, deserve to be shown. The seasonal variation of isoprene emissions (Figure 3) is evaluated against MEGAN-MACC and MEGAN-ERA5. What is the point of showing MEGAN-MACC? The seasonality should be evaluated against recent HCHO-based emission inversions. A part of the discrepancy between this study and MEGAN-ERA5 can be explained by the overestimation of emissions from Oceania. Still, Figure S12 suggests a large remaining bias even when removing Oceania. Is this due to differing seasonality in key emitting regions (e.g. Amazonia), or is it due to different geographical patterns?

**Response:**

We have revised Sections 3.5 and 3.6 to make them more concise and focused. Section 3.5 now concentrates on the regional contributions to global inter-phase variations, while Section 3.6 focuses on the analysis of key environmental drivers and includes some comparisons with the MEGAN-ERA5 inventory. Please refer to Section 3.5 and 3.6 for details.

To better validate the isoprene emissions seasonality in this study, we have conducted sensitivity inversions using MEGAN-MACC and MEGAN-ERA5 as isoprene prior in 2019, respectively, which shows a minimal difference (<3.5%) in global annual totals. Both MEGAN-MACC and MEGAN-ERA5 derived posteriors show a peak in JAS but reach minimum in DJF period, consistent with our findings. Besides, the satellite observed isoprene and HCHO column concentrations also exhibit a similar seasonal pattern as posteriors, which further demonstrates the reliability of isoprene posterior peak in JAS while minimum in DJF. Detailed discussion on

sensitivity inversion on prior has been added as Section 2.6 The impact of prior choice on inferred isoprene emissions in the main text, and the aligned seasonality of the recent HCHO-based isoprene inversion, different prior tests, and satellite observations have been added in Lines 363-366 and Lines 374-378.

**2.6 The impact of prior choice on inferred isoprene emissions**

To evaluate the sensitivity of the inversion to the choice of prior emissions, two additional sensitivity experiments are conducted using MEGAN-MACC (Sindelarova et al., 2014) and MEGAN-ERA5 (also known as CAMS-GLOB-BIOv3.1) (Sindelarova, 2021; Sindelarova et al., 2022) isoprene inventories, both of which are mechanistically distinct from the ORCHIDEE-based prior employed in the main analysis. The inversions are performed for the year 2019 following the same setup and observational constraints. Results show that the inferred global total isoprene emissions differ by less than 3.5% among the three prior configurations: deviations between the MEGAN-MACC-based inversion (500 Tg yr-1) and our posterior global total (485 Tg yr-1) are 3.1%, while those between the MEGAN-ERA5-based inversion (495 Tg yr-1) and our posterior are 2.1%, suggesting that the inversion framework remains robust to the choice of prior in global annual totals (Fig. S15). From a regional perspective, the largest differences occur in Oceania, where posterior emissions derived from MEGAN-MACC and MEGAN-ERA5 differ from our reference posterior by 60.6% and 17.4%, respectively (Fig. S16). Although Oceania shows the largest posterior discrepancies globally, these differences are substantially smaller than those in their priors (19 Tg yr-1 in ORCHIDEE, 108 Tg yr-1 in MEGAN-MACC, and 61 Tg yr-1 in MEGAN-ERA5 in 2019), indicating that the inversion effectively reconciles regional inconsistencies and converges toward observational constraints even where prior emissions diverge markedly. Overall, these tests demonstrate that the optimized emissions are primarily driven by observational constraints rather than by the characteristics of the prior inventory.

**Lines 363-366:**

"This seasonal cycle agrees with recent HCHO-based inversion results (Müller et al., 2024) but differs markedly from that in current bottom-up inventories: MEGAN-MACC (Sindelarova et al., 2014) and MEGAN-ERA5 (also known as CAMS-GLOB-BIOv3.1) (Sindelarova, 2021; Sindelarova et al., 2022) (Figs. 3 and S20)."

**Lines 374-378:**

"Besides, sensitivity inversions using MEGAN-MACC and MEGAN-ERA5 as priors also reproduce a JAS maximum and DJF minimum, reversing the original prior seasonality. The posterior seasonality derived from all three priors aligns with that observed in CrIS isoprene and OMPS HCHO concentrations (Fig. S20), indicating that the retrieved temporal variability reflects the observed atmospheric signals and demonstrating the robustness of the inferred seasonal cycle."

**Figure 3**. Monthly mean isoprene emissions from 2013 to 2020. (a) shows the global monthly pattern of ORCHIDEE prior and our posterior in this study, MEGAN-ERA5 (also known as CAMS-GLOB-BIOv3.1) inventory (Sindelarova, 2021) and posterior based on MEGAN-ERA5, as well as OMI HCHO-based isoprene inversion result (Müller et al., 2024). MEGAN-ERA5 is based on MEGAN v2.1, updated with ERA5 meteorology and CLM4 land cover (Sindelarova et al., 2022). (b)-(c) display monthly distributions of our estimated isoprene emissions (TgC) and temperature (K) by every 10° latitude band, respectively. We here only present the latitude range from 60°S to 60°N where emissions dominate (~99%). Temperature is acquired from ERA5. The monthly distributions of two MEGAN inventories (MEGAN-MACC and MEGAN-ERA5), precipitation from ERA5, and the Leaf area index (LAI) from Pu et al. (2024) are presented in Fig. S25.

**Figure S20.** Monthly variation of isoprene emissions (ORCHIDEE prior and posterior, MEGAN-MACC prior and posterior, MEGAN-ERA5 and posterior) and CrIS observed isoprene column and OMPS HCHO column concentrations.

**Minor comments**

l. 18: "introducing substantial uncertainties due to complex and nonlinear chemical pathways": wrong point to make, because the isoprene-based inversion is also subject to such uncertainties. The main "selling point" of CrIS-based inversion is of course the direct observation of isoprene, whereas formaldehyde is produced from the oxidation of many other VOCs. Please rephrase.

**Response:**

We have rephrased the original statement to "Most existing top-down atmospheric estimates of isoprene emissions rely on observational formaldehyde (HCHO) as an indirect proxy, even though HCHO is produced from multiple precursors." in Lines 17-18.

1. 34: Is precipitation really a driver of isoprene emissions? It is correlated with cloudiness and there anti-correlated with radiation. It also affects drought stress. The causes for correlation between top-down emissions and precipitation are therefore generally unclear. Rephrase, and adapt in the discussion.

**Response:**

Indeed, precipitation is not an explicit driver of biogenic isoprene emissions in process-based models such as MEGAN, but rather exerts indirect effects via changes in radiation and soil moisture that influence photosynthetic activity and drought stress. As these effects are already represented by the radiation and drought indicators (SPEI) in our driver analysis, we have removed analysis of precipitation as a driver and revised accordingly in the manuscript.

1. 43: Delete "precipitation"

**Response:**

**Done**

1. 47: add shrub to the land cover types

**Response:**

**Done**

1. 73-74: The main source of uncertainty might be that the emission factors for many plant species are currently unknown, e.g. over tropical forests.

**Response:**

We have added "unclear EFs especially over tropical regions" in Line 73.

1. 76: "spatial correlation": there is more than just correlation.

**Response:**

We have changed "spatial correlation" to "relationship" in Line 77.

1. 78-79: As explained above, isoprene concentrations are even more affected by non-linear chemistry than formaldehyde production rates.

**Response:**

We have added a supplementary explanation as "However, HCHO-based inversions face inherent limitations, including the non-linear nature of isoprene—OH chemistry (Valin et al., 2016) which is also a challenge for isoprene-based inversions" in Lines 78-79.

1. 79-80: "non-zero isoprene/HCHO lifetimes that smear the retrieved isoprene emissions": rephrase, unclear.

**Response:**

We have rephrased the original statement to "smearing effects causing spatial displacement between isoprene emissions and HCHO formation" in Lines 80-81.

1. 83: replace "potentially" by "partially".

**Response:**

**Done.**

1. 94-95: "overcoming limitations of traditional HCHO-based...": see above, rephrase, taking into account the isoprene-based inversions have their own limitations.

**Response:**

We have rephrased the original statement to "complementing traditional HCHO-based approaches" in Line 96.

1. 108: Are the monthly-mean model columns sampled as the CrIS observations (i.e. ignoring days when CrIS data are absent)?

**Response:**

Yes, we have aligned the daily concentration coverage and then calculate monthly-mean concentrations.

1. 125: What version of TROPOMI NO2 is used?

**Response:**

The TROPOMI NO2 we adopted is TROPOMI-RPRO-v2.4. We have added this information in Line 126.

Figure S3: the color bar is inadequate, please narrow it down and discuss potential differences with the corresponding distribution of beta from Wells et al. (2020) (their Figure S9).

**Response:**

We have narrowed the color bar range in original Fig. S3 (now Fig. S4) to 0.4-1.0, which follows a similar pattern of  $\beta$  from Wells et al. (2020). We have added some discussion in Lines 205-207.

**Lines 205-207:**

"Lower  $\beta$  values (around 0.6-0.7) are generally found over tropical hotspots such as the Amazon, while higher values ( $\geq 1$ ) are found across much of the Northern Hemisphere, similar to previous studies (Wells et al., 2020)."

**Figure S4**. An example of monthly  $\beta$  distribution in 2019.

1. 222 "indicating that real-world differences in beta are likely modest": this is absurd, the globally averaged difference is irrelevant.

**Response:**

We have rephrased the original statement to "In fact, emission variations are typically moderate; in this study, more than 63% of the grid cells exhibit posterior-prior differences within 65%, accounting for over 82% of the global total emissions on average, suggesting that  $\beta$  is relatively insensitive to the magnitude of emission perturbations in most regions (Fig. S9)." in Lines 225-228.

1. 229 "prior overestimation": rephrase. The overestimation is far from being ubiquitous.

**Response:**

We have rephrased original sentence to "reflecting a substantial improvement in model—observation agreement relative to the prior simulation" in Line 297.

1. 239: The validation using PGN data shows an almost negligible improvement. Note that the number and location of the PGN stations is not ideal for this validation. Since the number of stations steadily increases, consider using 2020 for this validation.

**Response:**

We have added an additional validation using PGN data for 2020, when more stations are available. The results show a similar improvement of the posterior relative to the prior, with slope increasing from 0.58 to 0.62 and the RMSE decreasing from  $0.49 \times 10^{16}$  to  $0.47 \times 10^{16}$  molecules cm-2. We have added this result in Lines 308-310 and Fig. S19.

**Lines 308-310:**

"In 2020, when more PGN sites became available (increasing from 15 in 2019 to 20), the posterior HCHO concentrations also better match the PGN observations, with the RMSE decreasing from  $0.49 \times 10^{16}$  to  $0.47 \times 10^{16}$  molecules cm-2 (Fig. S19)."

**Figure S19.** Comparison of HCHO column concentration between simulation and PGN surface observation in 2019 and 2020. (a) and (b) show the distribution of PGN stations used in this study, which provided official data within 60°S and 60°N for 2019 and 2020, respectively. (c) compares the correlation between posterior simulated, prior simulated, and PGN observed HCHO column concentrations in 2019 and 2020, respectively. PGN data are acquired from https://www.pandonia-global-network.org/.

Figure 1 is difficult to read due to the small size of the maps. The color bar leads to saturation in high-emission areas, while most other regions are very dark. Consider using a non-linear color scale to improve clarity.

**Response:**

We have replotted Figure 1 as shown below, using a logarithmic color scale.

Figure 1. Evaluation of the posterior LMDZ-INCA simulation using TROPOMI HCHO and CrIS isoprene observations in 2019. (a) and (b) present the comparison of the simulated HCHO with TROPOMI observations, and of the simulated isoprene with CrIS observations, respectively. From top to bottom: the global distribution of model grid-scale annual mean of the posterior simulation, satellite observation (from TROPOMI in (a) column and from CrIS in (b) column), prior simulation of the column concentrations, and correlation between annual-mean simulation and observation across the model grid-cells covered by the observation.

1. 283: the uncertainty of 43.8% for global emissions is not compatible with Figure 2(b), which shows values well below 40% everywhere (except >60N). Also, how can the uncertainty be so uniform in space, except for the lower values over high-column areas? Over low-column regions (e.g. deserts), one would expect uncertainties close to the prior (117%). Please clarify.

**Response:**

We previously used a uniform prior uncertainty and a three-segmented observation uncertainty, which resulted in a relatively uniform posterior distribution. We have now refined this by introducing a continuous linear scaling of uncertainty for low-column grids (below  $2\times10^{15}$  molec cm-2), interpolated from the  $2-10\times10^{15}$  molec cm-2 range and capped at 100%. This adjustment increases uncertainties over low-column regions, improving spatial consistency with expectations.

We have updated the uncertainty results throughout the manuscript, especially in Section 3.2. The uncertainty map has been updated in Fig. 2 as shown below:

Figure 2. (a) Global distribution of isoprene emissions (TgC per grid cell of 1.27° latitude × 2.5° longitude per year) and (b) relative uncertainties (%) in 2020. The uncertainties of global totals are area-weighted averages.

1. 300-301: "consistent with our posteriors": not so much, the seasonal profiles are still very different.

**Response:**

We have rephrased the original statement to "exhibiting a broadly similar seasonal pattern to our posteriors (Fig. S22)." in Lines 373-374.

1. 312-316: Such a long explanation... there is simply much more mid-latitude area in NH compared to SH.

**Response:**

We have refined this part to "Notably, the synchronicity between monthly emissions and temperature is stronger in the Northern Hemisphere (R=0.96) than in the Southern Hemisphere (R=0.54), reflecting the greater extent of mid-latitude land areas and sharper temperature seasonality in the north (Figs. 3b-3c, and S24). Additionally, stronger LAI variations in the Northern Hemisphere further reinforce this seasonal pattern (Figs. S25-S26) (Ren et al., 2024; Ma et al., 2023)." in Lines 387-391.

Figure 3: The precipitation subplot is not useful. The LAI subplot does not bring much either.

**Response:**

We have moved precipitation and LAI subplots to Figure S25, and re-plotted Figure 3 as shown below.

**Figure 3. Monthly mean isoprene emissions from 2013 to 2020.** (a) shows the global monthly pattern of ORCHIDEE prior and our posterior in this study, MEGAN-ERA5 (also known as CAMS-GLOB-BIOv3.1) inventory (Sindelarova, 2021) and posterior based on MEGAN-ERA5, as well as OMI HCHO-based isoprene inversion result (Müller et al., 2024). MEGAN-ERA5 is based on MEGAN v2.1, updated with ERA5 meteorology and CLM4 land cover (Sindelarova et al., 2022). (b)-(c) display monthly distributions of our estimated isoprene emissions (TgC) and temperature (K) by every 10° latitude band, respectively. We here only present the latitude range from 60°S to 60°N where emissions dominate (~99%). Temperature is acquired from ERA5. The monthly distributions of two MEGAN inventories (MEGAN-MACC and MEGAN-ERA5), precipitation from ERA5, and the Leaf area index (LAI) from Pu et al. (2024) are presented in Fig. S25.

Figure 4 (a) is not very clear, it is difficult to distinguish the lines.

**Response:**

We have re-plotted Figure 4 to make the lines more distinguishable, as shown below.

Canada (CAN) | Europe (EU) | Russia+Central Asia (RUS+CAS) | United States (USA) | Mideast (MIDE) China+Korea+Japan (CHN+KAJ) | South Asia (SAS) | Central America (CAM) | Northern Africa (NAF) | Amazon (AMZ) Rest of Southern America (RSAM) | Equatorial Africa (EQAF) | Southeast Asia (SEAS) | Southern Africa (SAF) | Oceania (OCE)

**Figure 4. Interannual isoprene emission variations from 2013 to 2020.** (a) compares the annual global isoprene emissions among the posterior (red shadow indicate the uncertainty), inventories including MEGAN-MACC, the MEGAN-ERA5 (also known as CAMS-GLOB-BIOv3.1) inventory, ensembles from Opacka et al. (2021), ensembles from CMIP6 (Do et al., 2025), and inversions based on corrected OMI HCHO observations (Müller et al., 2024). (b) plots the global spatial distribution of 1σ of annual isoprene emissions from 2013 to 2020, with frames corresponding to regions discussed in text. (c) depicts the regional annual emissions as well as the emission intensities (defined as the annual isoprene emissions per square meter per year). The regional classification is detailed in Fig. S6 of the SI and full names are listed below the figure.

Sections 3.5-3.6: I find that these sections should be shortened. Attribution of causes to the observed correlation is often speculative and uncertain, due to the co-variation of different factors.

**Response:**

We have revised Sections 3.5 and 3.6 to make them more concise and focused. Section 3.5 now concentrates on the regional contributions to global inter-phase variations, while Section 3.6 focuses on the analysis of key environmental drivers and includes some comparisons with the MEGAN-ERA5 inventory. Please refer to Section 3.5 and 3.6 for details.

1. 386: "amplified sensitivity" and 1. 390 "enhanced temperature sensitivity": rephrase. There are other factors than temperature. Only the apparent temperature sensitivity is enhanced, not the real one.

**Response:**

We have removed this discussion from Section 3.5 to keep the section focused on regional contributions to global variability.

l. 505: "with atypical vertical profiles": rephrase. Model have difficulties reproducing vertical profiles of short-lived species (not just in atypical situations). A discussion of this aspect would be needed, in light of model comparisons with aircraft data (for other species, from previous papers).

**Response:**

We have rephrased the original statement to "The ANN-based retrieval lacks scene-specific vertical sensitivity information, introducing additional uncertainty in aligning the vertical profiles between observations and the model." in Lines 556-558.

l. 515-516: the linearity clearly breaks down in many regions, not just at low NOx. This is shown by the posterior model overestimation of CrIS columns in many regions, as mentioned above.

**Response:**

We have rephrased this statement to "Nevertheless, the linearity between isoprene columns and emissions may break down across regions, especially in high-isoprene, low-NOx environment like the Amazon, where OH levels are limited (Zhao et al., 2025; Yoon, 2025)." in Lines 571-573.

Besides, discussions on the linearity have been detailed in Section 2.4 The linearity between isoprene concentrations and emissions.

Section 6: This section lacks substance. The "findings" (climate sensitivity of emissions, etc.) are not new. I fail to see what we really learned from the emission inversions. E.g., is T-sensitivity too weak or too strong in MEGAN in some regions? Where are biogenic emission models successful, and where do they fail?

**Response:**

We have added more comparisons between our posteriors and MEGAN-ERA5 inventory in Lines 500-506 in Section 3.6, and summarized the difference between posteriors and MEGAN-ERA5 inventory to highlight the findings in Lines 604-611 in Section 6. In short, we find similar positive correlations between isoprene emissions and temperature except in EQAF, and the biggest difference is the opposite seasonality of isoprene emissions between our inversion results and current MEGAN inventory.

Lines 500-506 in Section 3.6:

"Across most regions, isoprene emissions show strong positive correlations with temperature (R > 0.5, p < 0.05; Fig. 7a), suggesting temperature as the dominant first-order driver. Similar patterns are also observed in the MEGAN-ERA5 inventory (Fig. S36). However, a notable difference appears in EQAF, where our posterior results show no significant correlation with temperature, whereas MEGAN-ERA5 exhibits a strong positive correlation. This finding is consistent with previous HCHO-based isoprene inversion studies, which reported a reduced temperature dependence of isoprene emissions in the EQAF region (emission factor decreased from 4.3 to 2.7 for evergreen broadleaf trees) (Marais et al., 2014)."

**Lines 604-611 in Section 6:**

"This seasonal pattern contrasts with the JAS minimum and DJF peak simulated by the two MEGAN inventories. Sensitivity inversions using MEGAN-MACC and MEGAN-ERA5 as priors yield consistent posterior seasonality, suggesting that bottom-up inventories likely overestimate emissions in the Southern Hemisphere, especially over Oceania. Regarding temperature sensitivity, MEGAN-based emissions generally display a more uniform response to temperature, whereas our inversion indicates regionally differentiated sensitivities. For instance, in EQAF, temperature is not the apparent dominant driver, implying that other factors, such as vegetation dynamics or solar radiation, exert a stronger influence than represented in current models."

**Technical comments**

1. 26: replace "surface observations" by "ground-based optical measurements"

**Response:**

Done.

1. 168: and elsewhere: replace "low NO2" by "low NOx"

**Response:**

Done.

1. 170: replace NO2 by NOx

**Response:**

Done.

1. 195: Impact of NOx

**Response:**

Done.

Figure 1, 2, 4 and in the Supplement: why is Antarctica wrongly shaped? You could limit the plot to 60S - 90N.

**Response:**

We have replotted all maps within 60°S-90°N in Figure 1, 2 and 4 in the manuscript, and Figure S1, 3, 4, 6, 10, 12, 14, 15, 17, 19, and 26 in SI.

**References**

Abbot, D. S. et al., Geophys Res. Lett., 30, 1886, doi:10.1029/2003GL017336, 2003.

Folberth, G. A. et al., Atmos. Chem. Phys. 6, 2273, doi:10.5194/acp-6-2273-2006, 2006.

Lathiere, J. et al., Atmos. Chem. Phys., 6, 2129, https://doi.org/10.5194/acp-6-2129-2006, 2006.

Naik, V. et al., J. Geophys. Res., 109, D06301, doi:10.1029/2003JD004236, 2004

Novelli, A. et al., Atmos. Chem. Phys. 20, 3333, https://doi.org/10.5194/acp-20-3333-2020, 2020.

Stavrakou, T. et al., Geophys. Res. Lett. 45, 8681, https://doi.org/10.1029/2018GL078676, 2018.

Wells, K. C. et al., Nature 585, 225, https://doi.org/10.1038/s41586-020-2664-3, 2020.

Wennberg, P. O. et al., Chem Rev. 118, 3337, DOI: 10.1021/acs.chemrev.7b00439, 2018.

**Response:**

We have added all of these references at appropriate locations throughout the manuscript, as specified below.

**Lines 479-480:**

"The strong temperature sensitivity of USA isoprene emissions is consistent with previous study (Abbot et al., 2003)."

**Lines 156-160:**

"The mechanism comprises 398 homogeneous, 84 photolytic, and 33 heterogeneous reactions, and is continuously updated to integrate newly identified chemical processes and reaction pathways,

thereby improving the representation of atmospheric composition and oxidation capacity (Hauglustaine et al., 2004; Folberth et al., 2006; Pletzer et al., 2022; Sand et al., 2023; Terrenoire et al., 2022; Novelli et al., 2020; Wennberg et al., 2018)."

**Lines 600-602:**

"The elevated biogenic isoprene emissions during the El Niño period are consistent with previous studies (Lathière et al., 2006; Naik et al., 2004)."

**Lines 50-52:**

"Of all climate variables, temperature is widely recognized as the primary driver (Seco et al., 2022; Stavrakou et al., 2018), yet the variability of its influence across regions is not well characterized."

**Lines 205-207:**

"Lower  $\beta$  values (around 0.7) are generally found over tropical hotspots such as the Amazon, while higher values ( $\geq 1$ ) are observed across much of the Northern Hemisphere, consistent with previous studies (Wells et al., 2020)."

---

## Author Comment (AC3)

This study develops a global, multi-year inversion of biogenic isoprene emissions by assimilating CrIS-retrieved isoprene columns into the LMDZ-INCA chemistry—transport model, producing monthly emission estimates for 2013–2020. The approach represents a meaningful advancement in directly constraining isoprene emissions, distinct from traditional HCHO-based inversions. The manuscript is clearly structured, and the results are generally consistent with existing inventories, providing valuable insights into the spatial and interannual variability of global biogenic sources. Overall, I find the study scientifically relevant and well executed, but several methodological and interpretative aspects require clarification before publication.

**Response:**

We greatly appreciate the referee's valuable and perceptive feedback on our manuscript. Below, we provide detailed responses addressing each point raised.

**Model description and chemical mechanism**

A more detailed description of the atmospheric transport model (LMDZ-INCA) is needed, particularly the configuration of isoprene-related chemistry and its coupling with oxidants such as OH and NOx. Since the chemical mechanism largely governs how simulated isoprene columns respond to emission perturbations, the manuscript should specify the chemical mechanism (i.e., relevant reaction pathways and key rate constants). This will allow readers to assess the reliability and representativeness of the inversion results.

**Response:**

We have expanded the description of the VOC chemical mechanisms in the manuscript (Lines 153–160), with particular attention to isoprene and HCHO. Briefly, the LMDZ-INCA model features a comprehensive reactive VOC chemistry scheme that incorporates 14 reactions for isoprene and 80 for HCHO, based on the up-to-date reaction rates.

**Lines 153-160 in Section 2.2:**

"LMDZ-INCA contains a state-of-the-art CH4–NOx–CO–NMHC–O3 tropospheric photochemistry scheme with a total of 174 tracers, including the chemical degradation scheme of 10 non-methane hydrocarbons (NMHCs): C2H6, C3H8, C2H4, C3H6, C2H2, a lumped C>4 alkane, a lumped C>4 alkene, a lumped aromatic, isoprene and α-pinene. The mechanism comprises 398 homogeneous, 84 photolytic, and 33 heterogeneous reactions, and is continuously updated to integrate newly identified chemical processes and reaction pathways, thereby improving the representation of atmospheric composition and oxidation capacity (Hauglustaine et al., 2004; Folberth et al., 2006; Pletzer et al., 2022; Sand et al., 2023; Terrenoire et al., 2022; Novelli et al., 2020; Wennberg et al., 2018). Reactions directly related to isoprene and HCHO are listed in Tables S1-S2."

**Sensitivity to NOx levels**

Given that the oxidation rate of isoprene depends on ambient NOx conditions, it would be valuable to perform an additional sensitivity experiment under NO2 concentrations closer to TROPOMI observations.

**Response:**

We performed a sensitivity experiment by increasing  $NO_x$  emission inputs by 25% for the year 2019. This adjustment resulted in a closer agreement of simulated  $NO_2$  from LMDZ-INCA with TROPOMI observations, alleviating the original underestimation (Fig. S13). The global annual total from this inversion differs by less than 1% from the base inversion, with the largest regional deviation found in South Asia (SAS). The results of this sensitivity test have been added in Lines 270–273 in the manuscript and illustrated in Fig. S14.

Lines 270–273:

"To further assess the influence of  $NO_x$  conditions on the inversion, we perform a sensitivity test using +25%  $NO_x$  emissions for 2019. The results show negligible differences from the base inversion, with a global annual total deviation of less than 0.1% and the largest regional difference of 0.9% over South Asia (SAS) (Fig. S14)."

**Figure S14.** Comparison between base inversion and sensitivity inversion with  $\pm 25\%$  NOx emission input. (a) presents the global distribution of monthly difference in isoprene posteriors, and (b) compares the regional annual isoprene posteriors.

**Seasonality of posterior emissions**

The seasonality of posterior emissions in this study appears to differ from that of existing inventories, yet the underlying cause of this discrepancy remains unclear. It would be helpful to disentangle the contributions from the prior and the observational constraint. I suggest presenting the seasonal cycles of the prior (e.g., ORCHIDEE), satellite observations (CrIS isoprene and satellite HCHO), and possibly a parallel inversion using MEGAN as the prior. Comparing these seasonal patterns would clarify whether the retrieved seasonality reflects model, data adjustments, or inherent observational features, thereby strengthening the validation of the inferred isoprene emission seasonality.

**Response:**

To evaluate the robustness of the seasonal pattern identified in this study, we conducted sensitivity inversions for 2019 using MEGAN-MACC and MEGAN-ERA5 as alternative isoprene priors. Both inversions produce posterior emissions that peak during JAS and reach a minimum in DJF, consistent with our posteriors. In addition, satellite-observed isoprene and HCHO column concentrations display similar seasonal variations, supporting the reliability of the posterior peak in JAS and minimum in DJF. These consistent seasonal features across the HCHO-based isoprene inversion, prior sensitivity tests, and satellite observations have been incorporated in Lines 363–366, 374–378, Figure 3, and Figure S20.

**Lines 363-366:**

"This seasonal cycle agrees with recent HCHO-based inversion results (Müller et al., 2024) but differs markedly from that in current bottom-up inventories: MEGAN-MACC (Sindelarova et al., 2014) and MEGAN-ERA5 (also known as CAMS-GLOB-BIOv3.1) (Sindelarova, 2021; Sindelarova et al., 2022) (Figs. 3 and S20)."

**Lines 374-378:**

"Besides, sensitivity inversions using MEGAN-MACC and MEGAN-ERA5 as priors also reproduce a JAS maximum and DJF minimum, reversing the original prior seasonality. The posterior seasonality derived from all three priors aligns with that observed in CrIS isoprene and OMPS HCHO concentrations (Fig. S20), indicating that the retrieved temporal variability reflects the observed atmospheric signals and demonstrating the robustness of the inferred seasonal cycle."

**Figure 3**. Monthly mean isoprene emissions from 2013 to 2020. (a) shows the global monthly pattern of ORCHIDEE prior and our posterior in this study, MEGAN-ERA5 (also known as CAMS-GLOB-BIOv3.1) inventory (Sindelarova, 2021) and posterior based on MEGAN-ERA5, as well as OMI HCHO-based isoprene inversion result (Müller et al., 2024). MEGAN-ERA5 is based on MEGAN v2.1, updated with ERA5 meteorology and CLM4 land cover (Sindelarova et al., 2022).

(b)-(c) display monthly distributions of our estimated isoprene emissions (TgC) and temperature (K) by every 10° latitude band, respectively. We here only present the latitude range from 60°S to 60°N where emissions dominate (~99%). Temperature is acquired from ERA5. The monthly distributions of two MEGAN inventories (MEGAN-MACC and MEGAN-ERA5), precipitation from ERA5, and the Leaf area index (LAI) from Pu et al. (2024) are presented in Fig. S25.

**Figure S20.** Monthly variation of isoprene emissions (ORCHIDEE prior and posterior, MEGAN-MACC prior and posterior, MEGAN-ERA5 and posterior) and CrIS observed isoprene column and OMPS HCHO column concentrations.

**Regional contribution and structure of Sections 3.5–3.6**

Section 3.5 could be more focused on quantifying and interpreting regional contributions to global variability, while Section 3.6 emphasizes the environmental drivers. This clearer separation would improve the logical flow and avoid redundancy between the two sections. Additionally, including comparisons with MEGAN or other inventories in the driver analysis (Section 3.6) would contextualize the inversion results and highlight the added value of the CrIS-based framework in capturing regional differences and environmental controls on isoprene emissions.

**Response:**

We have streamlined Sections 3.5 and 3.6 to make the discussion more concise and focused. Section 3.5 now highlights the regional contributions to global inter-phase variations, whereas Section 3.6 concentrates on the analysis of key environmental drivers and includes comparisons with the MEGAN-ERA5 inventory. Please refer to Section 3.5 and 3.6 for details.

**Specific comments:**

Line 125: Please specify the TROPOMI NO2 version.

**Response:**

**The TROPOMI NO2 we adopted is TROPOMI-RPRO-v2.4. We have added this information in Line 126.**

Section 3.2: In the current uncertainty configuration, the authors apply a simplified three-segment scheme for the CrIS observational uncertainty, assigning a uniform 70% uncertainty for all low-column grids. This assumption may underestimate the uncertainty gradient in regions with very low isoprene concentrations. It is recommended to refine this setting by introducing a continuous or extended scaling below the lower threshold, so that the relative uncertainty increases progressively as the concentration decreases.

**Response:**

We have modified our uncertainty settings as suggested. Specifically, we introduce a continuous linear scaling of uncertainty for low-column grids (below  $2\times10^{15}$  molec cm-2), interpolated from the  $2-10\times10^{15}$  molec cm-2 range and capped at 100%. This adjustment increases uncertainties over low-column regions, improving spatial consistency with expectations. We have updated the uncertainty results throughout the manuscript, especially in Section 3.2. The uncertainty map has been updated in Fig. 2 as shown below:

Figure 2. (a) Global distribution of isoprene emissions (TgC per grid cell of 1.27° latitude × 2.5° longitude per year) and (b) relative uncertainties (%) in 2020. The uncertainties of global totals are area-weighted averages.

Section 3.3: To validate the seasonality of the posterior emissions, comparison with inventories alone is insufficient. I suggest that the authors include HCHO-based inversion results as an additional reference.

**Response:**

We have added an OMI HCHO-based isoprene inversion result (Müller et al., 2024) in Figure 3 (yellow curve), which show a similar isoprene emission seasonality to our posteriors. We have added this comparison in Lines 363-366.

**Lines 363-366:**

"This seasonal cycle agrees with recent HCHO-based inversion results (Müller et al., 2024) but differs markedly from that in current bottom-up inventories: MEGAN-MACC (Sindelarova et al., 2014) and MEGAN-ERA5 (also known as CAMS-GLOB-BIOv3.1) (Sindelarova, 2021; Sindelarova et al., 2022) (Figs. 3 and S20)."

Table S3: The left parenthesis is missing.

**Response:**

We have complemented the left parenthesis.

Line 301: The use of "consistent" is not precise here, as the results are not actually identical or in full agreement in Fig. S12, please rephrase.

**Response:**

We have rephrased the original statement to "exhibiting a broadly similar seasonal pattern to our posteriors" in Lines 373-374.